# Learning from Historical Activations in Graph Neural Networks

**Yaniv Galron**
Technion – Israel Institute of Technology
yaniv.galron@campus.technion.ac.il

**Hadar Sinai**
Technion – Israel Institute of Technology
hadarsi@campus.technion.ac.il

**Haggai Maron**
Technion – Israel Institute of Technology
NVIDIA
haggaimaron@technion.ac.il

**Moshe Eliasof**
Ben-Gurion University of the Negev
University of Cambridge
eliasof@bgu.ac.il

## Abstract

Graph Neural Networks (GNNs) have demonstrated remarkable success in various domains such as social networks, molecular chemistry, and more. A crucial component of GNNs is the pooling procedure, in which the node features calculated by the model are combined to form an informative final descriptor to be used for the downstream task. However, previous graph pooling schemes rely on the last GNN layer features as an input to the pooling or classifier layers, potentially under-utilizing important activations of previous layers produced during the forward pass of the model, which we regard as *historical graph activations*. This gap is particularly pronounced in cases where a node's representation can shift significantly over the course of many graph neural layers, and worsened by graph-specific challenges such as over-smoothing in deep architectures. To bridge this gap, we introduce HISTOGRAPH, a novel two-stage attention-based final aggregation layer that first applies a unified layer-wise attention over intermediate activations, followed by node-wise attention. By modeling the evolution of node representations across layers, our HISTOGRAPH leverages both the activation history of nodes and the graph structure to refine features used for final prediction. Empirical results on multiple graph classification benchmarks demonstrate that HISTOGRAPH offers strong performance that consistently improves traditional techniques, with particularly strong robustness in deep GNNs. Our code is at https://github.com/YanivDorGalron/HISTOGRAPH.

## 1 Introduction

Graph Neural Networks (GNNs) have achieved strong results on graph-structured tasks, including molecular property prediction and recommendation (Ma et al., 2019; Gilmer et al., 2017; Hamilton et al., 2017). Recent advances span expressive layers (Maron et al., 2019; Morris et al., 2023; Frasca et al., 2022; Zhang et al., 2023a;b; Puny et al., 2023), positional and structural encodings (Dwivedi et al., 2023; Rampášek et al., 2022; Eliasof et al., 2023a; Belkin & Niyogi, 2003; Maskey et al., 2022; Lim et al., 2023; Huang et al., 2024), and pooling (Ying et al., 2018; Lee et al., 2019; Bianchi et al., 2020; Wang et al., 2020; Vinyals et al., 2015; Zhang et al., 2018; Gao & Ji, 2019; Ranjan et al., 2020; Yuan & Ji, 2020). However, pooling layers still underuse intermediate activations produced during message passing, limiting their ability to capture long-range dependencies and hierarchical patterns (Alon & Yahav, 2020; Li et al., 2019; Xu et al., 2019).

In GNNs, layers capture multiple scales: early layers model local neighborhoods and motifs, while deeper layers encode global patterns (communities, long-range dependencies, topological roles) (Xu et al., 2019), mirroring CNNs where shallow layers detect edges/textures and deeper layers capture object semantics (Zeiler & Fergus, 2014). Greater depth can overwrite early information (Li et al., 2018; Eliasof et al., 2022) and cause over-smoothing, making node representations indistinguishable (Cai & Wang, 2020; Nt & Maehara, 2019; Rusch et al., 2023). We address this by leveraging

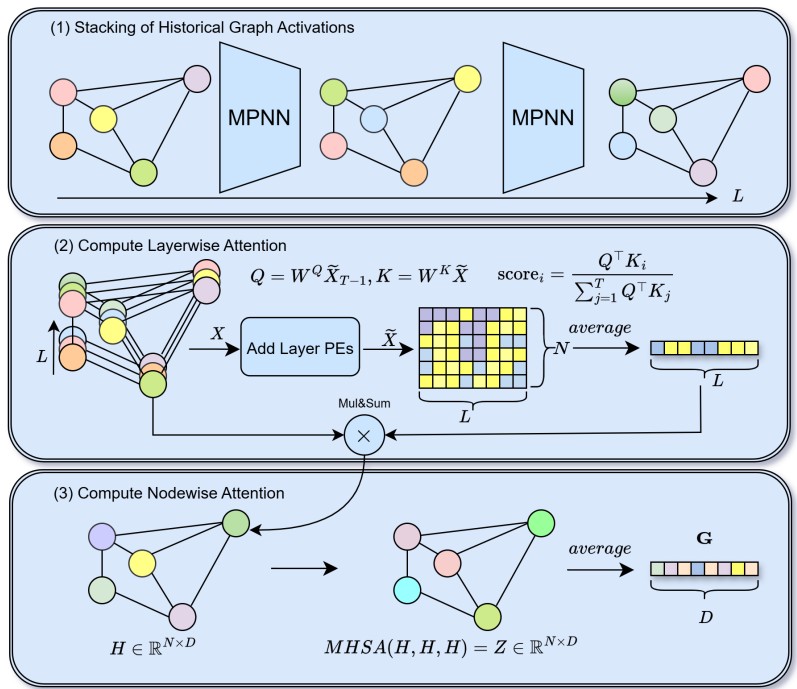

Figure 1: Overview of HISTOGRAPH. (1) Given input node features $\mathbf{X}_0$ and adjacency $\mathbf{A}$, a backbone GNN produces *historical graph activations* $\mathbf{X}_1, .., \mathbf{X}_{L-1}$. (2) The Layer-wise attention module uses the final-layer embedding as a query to attend over all historical states while averaging across nodes, yielding per-node aggregated embeddings $\mathbf{H}$. (3) A Node-wise self-attention module refines $\mathbf{H}$ by modeling interactions across nodes, producing $\mathbf{Z}$, then averaged if graph embeddings $\mathbf{G}$ is wanted.

*historical graph activations*, the representations from all layers, to integrate multi-scale features at readout (Xu et al., 2018).

Several works have explored the importance of deeper representations, residual connections, and expressive aggregation mechanisms to overcome such limitations (Xu et al., 2018; Li et al., 2021; Bresson & Laurent, 2017). Close to our approach are specialized methods like state space (Ceni et al., 2025) and autoregressive moving average (Eliasof et al., 2025) models on graphs, that consider a sequence of node features obtained by initialization techniques. Yet, these efforts often focus on improving stability during training, without explicitly modeling the internal trajectory of node features across layers. That is, we argue that a GNN's computation path and the sequence of node features through layers can be a valuable signal. By reflecting on this trajectory, models can better understand which transformations were beneficial and refine their final predictions accordingly.

In this work, we propose HISTOGRAPH, a self-reflective architectural paradigm that enables GNNs to reason about their *historical graph activations*. HISTOGRAPH introduces a two-stage self-attention mechanism that disentangles and models two critical axes of GNN behavior: the evolution of node embeddings through layers, and their spatial interactions across the graph. The layer-wise module treats each node's layer representations as a sequence and learns to attend to the most informative representation, while the node-wise module aggregates global context to form richer, context-aware outputs. HISTOGRAPH design enables learning representations without modifying the underlying GNN architecture, leveraging the rich information encoded in intermediate representations to enhance many graph related predictions (graph classification, node classification and link prediction).

We apply HISTOGRAPH in two complementary modes: (1) end-to-end joint training with the backbone, and (2) post-processing as a lightweight head on a frozen pretrained GNN. The end-to-end variant enriches intermediate representations, while the post-processing variant trains only the head, yielding substantial gains with minimal overhead. HISTOGRAPH consistently outperforms strong GNN and pooling baselines on TU and OGB benchmarks (Morris et al., 2020; Hu et al., 2020),

demonstrating that computational history is a powerful, general inductive bias. Figure 1 overviews HISTOGRAPH.

**Main contributions.** (1) We introduce a self-reflective architectural paradigm for GNNs that leverages the full trajectory of node embeddings across layers; (2) We propose HISTOGRAPH, a two-stage self-attention mechanism that disentangles the layer-wise node embeddings evolution and spatial aggregation of node features; (3) We empirically validate HISTOGRAPH on graph-level classification, node classification and link prediction tasks, demonstrating consistent improvements over state-of-the-art baselines; and, (4) We show that HISTOGRAPH can be employed as a post-processing tool to further enhance performance of models trained with standard graph pooling layers.

## 2 RELATED WORKS

**Graph Neural Networks.** GNNs propagate and aggregate messages along edges to produce node embeddings that capture local structure and features (Scarselli et al., 2008; Gilmer et al., 2017). GNN architectures are typically divided into two families: spectral GNNs, defining convolutions with the graph Laplacian (e.g., ChebNet (Defferrard et al., 2016), GCN (Kipf & Welling, 2016)), and spatial GNNs, aggregating neighborhoods directly (e.g., GraphSAGE (Hamilton et al., 2017), GAT (Veličković et al., 2017)). Greater GNN depth expands receptive fields but introduces over-smoothing (Cai & Wang, 2020; Nt & Maehara, 2019; Rusch et al., 2023; Li et al., 2018) and over-

Table 1: Comparison of pooling methods based on intermediate representation usage, structural considerations, and layer-node modeling.

| Method | Int. Repr. | Struct. | Layer-Node Model. |
|---|---|---|---|
| JKNet (Xu et al., 2018) | **Yes** | No | No |
| Set2Set (Vinyals et al., 2015) | No | **Yes** | No |
| SAGPool (Lee et al., 2019) | No | **Yes** | No |
| DiffPool (Ying et al., 2018) | No | **Yes** | No |
| SSRead (Lee et al., 2021) | No | **Yes** | No |
| DKEPool (Chen et al., 2023) | No | **Yes** | No |
| SOPool (Wang & Ji, 2023) | No | **Yes** | No |
| GMT (Baek et al., 2021) | No | **Yes** | No |
| Mean/Max/Sum Pool | No | No | No |
| HISTOGRAPH (Ours) | **Yes** | **Yes** | **Yes** |

squashing (Alon & Yahav, 2020). Mitigations include residual and skip connections (Chen et al., 2020; Xu et al., 2018), graph rewiring (Topping et al., 2021), and global context via positional encodings or attention (Graphormer (Ying et al., 2021), GraphGPS (Rampášek et al., 2022)). Several models preserve multi-hop information for robustness and expressivity. HISTOGRAPH maintains node-embedding histories across propagation and fuses them at readout. Unlike per-layer mixing, this yields a consolidated multi-scale summary, mitigating intermediate feature degradation and retaining local and long-range information.

**Pooling in Graph Learning.** Graph-level tasks (e.g., molecular property prediction, graph classification) require a fixed-size summary of node embeddings. Early GNNs used permutation-invariant readouts such as sum, mean, and max (Gilmer et al., 2017; Zaheer et al., 2017), as in GIN (Xu et al., 2019). Richer structure motivated learned pooling: SortPool sorts embeddings and selects top-$k$ (Zhang et al., 2018); DiffPool learns soft clusters for hierarchical coarsening (Ying et al., 2018); SAGPool scores nodes and retains a subset (Lee et al., 2019). Set2Set uses LSTM attention for iterative readout (Vinyals et al., 2015), while GMT uses multi-head attention for pairwise interactions (Baek et al., 2021). SOPool adds covariance-style statistics (Wang & Ji, 2023). Concurrently, Wang et al. (2024) propose leveraging global interactive patterns for cross-graph interpretability, which targets a complementary goal to our per-graph readout. A recent survey (Liu et al., 2022) reviews flat and hierarchical techniques on TU and OGB benchmarks. Hierarchical approaches (e.g., Graph U-Net (Gao & Ji, 2019)) capture multi-scale structure but add complexity and risk information loss. In contrast, HISTOGRAPH directly pools historical activations: layer-wise attention fuses multi-depth features, node-wise attention models spatial dependencies, and normalization stabilizes contributions. This preserves information across propagation depths without clustering or node dropping. Table 1 summarizes design choices and shows HISTOGRAPH is the only method combining intermediate representations with structural information.

**Residual Connections.** Residuals are pivotal for deep GNNs and multi-scale features. Jumping Knowledge flexibly combines layers (Xu et al., 2019), APPNP uses personalized PageRank to preserve long-range signals (Gasteiger et al., 2018), and GCNII adds initial residuals and identity

mappings for stability (Chen et al., 2020). In pooling, Graph U-Net links encoder–decoder via skips (Gao & Ji, 2019), and DiffPool's cluster assignments act as soft residuals preserving early-layer information (Ying et al., 2018). Other methods show that learnable residual connections can mitigate oversmoothing (Eliasof et al., 2023b), and allow a dynamical system perspective on graphs (Eliasof et al., 2024). Differently, our HISTOGRAPH departs by introducing *historical pooling*: at readout, it accumulates node histories across layers, creating a global shortcut at aggregation that revisits and integrates multi-hop features into the final representation unlike prior models that apply residuals only within node updates or via hierarchical coarsening.

## 3 LEARNING FROM HISTORICAL GRAPH ACTIVATIONS

We introduce HISTOGRAPH, a learnable pooling operator that improves graph representation learning across downstream tasks by integrating layer evolution and spatial interactions in an end-to-end differentiable framework. Unlike pooling that operates on the last GNN layer, HISTOGRAPH treats hidden representations as a sequence of historical activations. It computes node embeddings by querying each node's history with its final-layer representation, then applies spatial self-attention to produce a fixed-size graph representation. Details appear in Appendix B and Algorithm 1; Figure 1 overviews HISTOGRAPH, and Table 1 compares to other methods.

**Notations.** Let $\mathbf{F} \in \mathbb{R}^{N \times D_{\text{in}}}$ denote the raw input node features, where $N$ is the number of nodes in the batch and $D_{\text{in}}$ is the input feature dimensionality. The initial representation is given by $\mathbf{X}^{(0)} = \text{Emb}_{\text{in}}(\mathbf{F})$, where $\text{Emb}_{\text{in}}$ is a linear layer projecting input features to the GNN hidden dimension. For each subsequent layer $l = 1, \ldots, L-1$, the representations are computed recursively as $\mathbf{X}^{(l)} = \text{GNN}^{(l)}(\mathbf{X}^{(l-1)})$, where $\text{GNN}^{(l)}$ denotes the $l$-th GNN layer.

We denote by $\mathbf{X} = [\mathbf{X}^{(0)}, \mathbf{X}^{(1)}, \ldots, \mathbf{X}^{(L-1)}] \in \mathbb{R}^{N \times L \times D}$ the *historical graph activations*, i.e., the stacked node embeddings across all $L$ layers. Each node thus has $L$ historical embeddings corresponding to different depths of message passing. When GNN layers produce activations with varying dimensionalities $D_0, D_1, \ldots, D_{L-1}$, we apply per-layer linear projections $W^{(l)} : \mathbb{R}^{D_l} \to \mathbb{R}^D$ to map each layer's output to a common hidden dimension $D$ before stacking into $\mathbf{X}$.

**Input Projection and Per-Layer Positional Encoding.** We project the historical activations to a common hidden dimension $D$ using a linear transformation:

$$\mathbf{X}' = \text{Emb}_{\text{hist}}(\mathbf{X}) \in \mathbb{R}^{N \times L \times D}. \tag{1}$$

To encode layer ordering, we add fixed sinusoidal positional encodings as in Vaswani et al. (2017):

$$P_{l,2k} = \sin\left(\frac{l}{10000^{2k/D}}\right), \quad P_{l,2k+1} = \cos\left(\frac{l}{10000^{2k/D}}\right), \tag{2}$$

for $0 \le l < L$, $0 \le k < D/2$, resulting in $\mathbf{P} \in \mathbb{R}^{L \times D}$. The positional encoding is broadcast across the node dimension and added element-wise to obtain layer-aware features: $\widetilde{\mathbf{X}}_{v,l} = \mathbf{X}'_{v,l} + \mathbf{P}_l$, for each node $v$ and layer $l$, yielding $\widetilde{\mathbf{X}} \in \mathbb{R}^{N \times L \times D}$.

**Layer-wise Attention and Node-wise Attention.** We view each node through its sequence of historical activations and use attention to learn which activations are most relevant. We use only the last-layer embedding as the query to attend over all historical states:

$$\mathbf{Q} = \widetilde{\mathbf{X}}_{L-1} W^Q \in \mathbb{R}^{N \times 1 \times D}, \quad \mathbf{K} = \widetilde{\mathbf{X}} W^K \in \mathbb{R}^{N \times L \times D}, \quad \mathbf{V} = \widetilde{\mathbf{X}} \in \mathbb{R}^{N \times L \times D}. \tag{3}$$

We apply scaled dot-product attention and average across nodes, obtaining a layer weighting scheme:

$$\mathbf{c} = \text{Average}\left(\frac{\mathbf{Q}\mathbf{K}^\top}{\sqrt{D}}\right) \in \mathbb{R}^{1 \times L}. \tag{4}$$

Rather than softmax, which enforces non-negative weights and suppresses negative differences, we apply a normalization that permits signed contributions $\alpha_l = \frac{c_l}{\sum_{l'=0}^{L-1} c_{l'}}$. This allows the model to

express additive or subtractive relationships between layers, akin to finite-difference approximations in dynamical systems. The cross-layer pooled node embeddings are computed as:

$$\mathbf{H} = \sum_{l=0}^{L-1} \alpha_l \cdot \widetilde{\mathbf{X}}_l = \sum_{l=0}^{L-1} \frac{c_l}{\sum_{l'=0}^{L-1} c_{l'}} \cdot \widetilde{\mathbf{X}}_l \quad \in \mathbb{R}^{N \times D}. \tag{5}$$

**Graph-level Representation.** We first aggregate each node's history weighted by relevance to the final state, with a residual recency bias from the final-layer query, into $\mathbf{H}$. To obtain a graph-level representation, we apply multi-head self-attention across nodes *exactly once at the readout stage*—not at every GNN depth—omitting spatial positional encodings to preserve permutation invariance:

$$\mathbf{Z} = \mathrm{MHSA}(\mathbf{H}, \mathbf{H}, \mathbf{H}) \in \mathbb{R}^{N \times D}, \tag{6}$$

optionally followed by residual connections and LayerNorm. Averaging over nodes yields $\mathbf{G} = \mathrm{Average}(\mathbf{Z}) \in \mathbb{R}^D$, which then feeds the final prediction head (typically an MLP). Crucially, because node-wise self-attention is applied only once at the final readout—rather than at every message-passing layer—it does not contribute to over-smoothing during the GNN forward pass and incurs only a single $O(N^2 D)$ cost. Early message-passing layers capture local interactions, whereas deeper layers encode global ones (Gasteiger et al., 2019; Chien et al., 2020). By attending across layers and nodes, HISTOGRAPH fuses local and global cues, retaining multi-scale structure and validating our motivation.

**Computational Complexity.** Layer-wise attention costs $O(LD)$ per node; spatial attention over $N$ nodes costs $O(N^2 D)$. Thus the per-graph complexity is

$$O(NLD + N^2 D) = O(N(L + N)D), \tag{7}$$

with memory $O(L + N^2)$ from attention maps. A naïve joint node–layer attention costs $O(L^2 N^2 D)$, which is prohibitive. Our two-stage scheme—first across layers ($O(LD)$ per node), then across nodes ($O(N^2 D)$)—avoids this. Since $L \ll N$ in practice, the dominant cost is $O(N^2 D)$, matching a *single* graph-transformer layer. A standard graph transformer that stacks $L$ such layers incurs $O(LN^2 D)$ (Yun et al., 2019); HISTOGRAPH reduces the depth coefficient from $L$ to 1, because layer-wise attention operates per node in $O(LD)$ and the $O(N^2 D)$ spatial attention is applied only once at readout. This decomposition keeps historical activations tractable despite the quadratic node term. Empirically, HISTOGRAPH adds only a slight runtime over a standard GNN forward pass (Figure 4) while delivering significant gains across multiple benchmarks, as seen in Tables 2, 3 and 17.

**Frozen Backbone Efficiency.** With a pretrained, frozen message-passing backbone, we train only the HISTOGRAPH head. We cache the $N \times L \times D$ activations per graph in one forward pass and skip gradients through the backbone, removing $O(L(ED + ND^2))$ work (where $E$ is the number of edges). The backward pass applies only to the head, $O(N(L + N)D)$, substantially reducing memory and training time. This is especially useful in low-resource or few-shot regimes, and when fine-tuning large datasets where repeated backpropagation through $L$ GNN layers is prohibitive.

**Scalability Considerations.** The current design is best suited for small-to-medium-sized graphs; we discuss scaling strategies for larger graphs in Appendix G.

## 4 PROPERTIES OF HISTOGRAPH

In this section, we discuss the properties of our HISTOGRAPH, which motivate its architectural design choices. While the general idea of attention over a sequence of representations is not unique to HISTOGRAPH, the specific combination with GNN historical activations yields properties that do not arise in standard attention-based pooling. In particular, these properties show how (i) layer-wise attention mitigates over-smoothing and acts as a dynamic trajectory filter, (ii) the signed normalization (rather than softmax) enables the architecture to approximate low/high pass filters over the layer trajectory, and (iii) node-wise attention at readout is beneficial in our HISTOGRAPH.

**HISTOGRAPH can mitigate Over-smoothing.** One key property of HISTOGRAPH is its ability to mitigate the over-smoothing problem in a simple way. As node embeddings tend to become

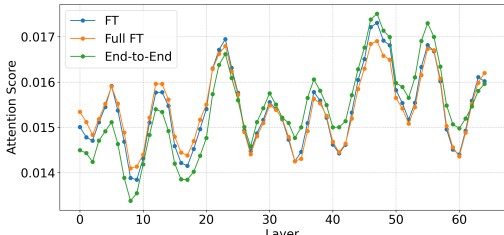 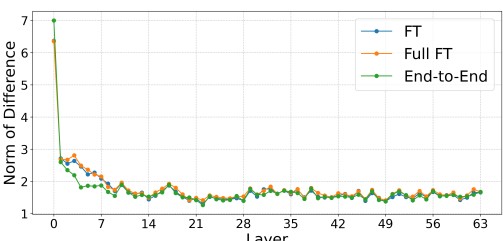

Figure 2: Visualizations on the IMDB-B dataset with 64-layer HISTOGRAPH. (left) Attention patterns across layers under different training regimes. (right) Embedding evolution throughout training, measured by the normed difference between final and intermediate representations.

indistinguishable after a certain depth $l_{os}$, i.e., $|\mathbf{x}_v^{(l_1)} - \mathbf{x}_u^{(l_2)}| = 0$ for all node pairs $u, v$ and layers $l_1, l_2 \geq l_{os}$, HISTOGRAPH aggregates representations across layers using a weighted combination:

$$\mathbf{h_u} = \sum_{l=0}^{L-1} \alpha_l \mathbf{x_u}^{(l)}, \quad \text{with} \quad \sum_l \alpha_l = 1. \tag{8}$$

Attention weights $\alpha_l$ that place nonzero mass on early layers let the final embedding $\mathbf{h}_u$ retain discriminative early representations, countering over-smoothing so node embeddings remain distinguishable ($|h_u - h_v| \neq 0$). This mechanism underlies HISTOGRAPH's robustness in deep GNNs, corroborated by the depth ablation in Table 17, Fig. 2 (which shows substantial early-layer attention and nonzero differences between historical activations), and the feature distance diagnostics in Table 12 (which confirms that HISTOGRAPH increases embedding diversity across all layers compared to a standard GCN). We formalize HISTOGRAPH's mitigation of over-smoothing in Proposition 1; the proof appears in Appendix E.

**Proposition 1** (Mitigating Over-smoothing with HISTOGRAPH). *Let $\mathbf{x}_v^{(l)} \in \mathbb{R}^D$ denote the embedding of node $v$ at layer $l$ of a GNN. Suppose the GNN exhibits over-smoothing, i.e., there exists some layer $L_0$ sufficiently large such that for all layers $l_1, l_2 > L_0$ and all nodes $u, v$,*

$$\|\mathbf{x}_u^{(l_1)} - \mathbf{x}_v^{(l_2)}\| \to 0. \tag{9}$$

*Let* HISTOGRAPH *compute the final node embedding as*

$$h_v = \sum_{l=0}^{L-1} \alpha_l \mathbf{x}_v^{(l)}, \tag{10}$$

*where $\alpha_l$ are learned attention weights. Then, for distinct nodes $u$ and $v$, there exists at least one layer $l' \leq L_0$ with $\alpha_{l'} \neq 0$ such that*

$$\|h_u - h_v\| > 0. \tag{11}$$

*That is,* HISTOGRAPH *retains information from early layers and mitigates over-smoothing.*

**Interpretability of Learned Attention Weights.** Figure 2 shows that HISTOGRAPH learns nontrivial layer weightings: substantial weight is placed on early (pre-over-smoothing) layers and the final layer, forming a task-adapted profile that balances local and global information. A detailed analysis appears in Appendix H.

**HISTOGRAPH's Layer-wise Attention is an Adaptive Trajectory Filter.** We interpret HISTOGRAPH's layer-wise attention as an Adaptive Trajectory Filter, which dynamically aggregates a node's embeddings across layers based on learned weights. Let $\{\mathbf{x}^{(l)}\}_{l=0}^{L-1} \subset \mathbb{R}^D$ denote a node's embeddings at each layer. We define the aggregated embedding as:

$$\mathbf{h} = \sum_{l=0}^{L-1} \alpha_l \mathbf{x}^{(l)}, \quad \text{with} \quad \sum_l \alpha_l = 1. \tag{12}$$

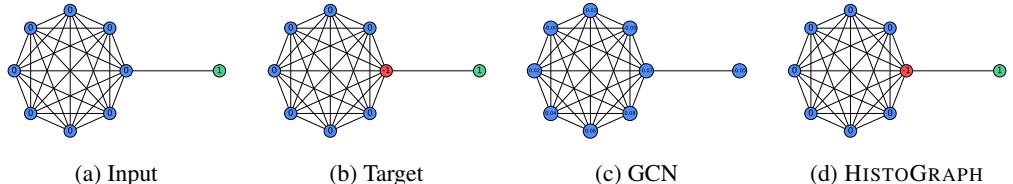

| (a) Input | (b) Target | (c) GCN | (d) HISTOGRAPH |

Figure 3: Graph and signal transformations: (a) input node features; (b) prediction target, the node-feature gradient; (c) GCN output trained to approximate (b) from (a); (d) HISTOGRAPH output. The gap between GCN and HISTOGRAPH underscores the importance of adaptive trajectory filtering. Node colors: red, blue, and green denote values $-1, 0, 1$.

where $\alpha_l$ are learnable attention weights. In general, any weighted combination over a sequence can be viewed as a filter. However, two design choices make HISTOGRAPH's filtering distinct from standard attention mechanisms. First, the *signed normalization* (division by sum rather than softmax) permits negative weights, enabling the model to express subtractive relationships between layers—analogous to a finite impulse response (FIR) filter with signed coefficients. This allows the aggregation to implement: (i) a low-pass filter when $\alpha_l = \frac{1}{L}$ (uniform average); (ii) a high-pass filter when $\alpha_l = \delta_{l,L-1} - \delta_{l,L-2}$ (first difference); and (iii) a general FIR filter when $\alpha_l$ are learned. Standard softmax-based attention, by contrast, is restricted to non-negative convex combinations and cannot realize subtractive (high-pass) filtering. Second, HISTOGRAPH applies this filtering specifically to the *GNN's computational trajectory*—a sequence of representations that progressively encode larger neighborhoods—rather than to an arbitrary set of features. This means the filter directly controls the balance between local (shallow-layer) and global (deep-layer) information at readout, a property that is specific to the GNN setting.

Figure 3 illustrates a case where GCN fails at high-pass filtering, whereas HISTOGRAPH succeeds. The barbell graph—a symmetric clique joined by a single edge—creates a sharp gradient discontinuity, highlighting how the adaptive filtering of HISTOGRAPH preserves such signals, unlike standard GCNs. Appendix D further analyzes the usefulness of node-wise attention in HISTOGRAPH.

## 5 EXPERIMENTS

In this section, we conduct an extensive set of experiments to demonstrate the effectiveness of HISTOGRAPH as a graph pooling function. Our experiments seek to address the following questions:

(Q1) Does HISTOGRAPH consistently improve GNN performance over existing pooling functions across diverse domains?

(Q2) Can HISTOGRAPH be applied as a post-processing step to enhance the performance of pretrained GNNs?

(Q3) What is the impact of each component of HISTOGRAPH on performance?

**Baselines.** We compare HISTOGRAPH against diverse baselines spanning graph representation and pooling. Message-passing GNNs: GCN and GIN with mean or sum aggregation (Kipf & Welling, 2016; Xu et al., 2019). Set-level pooling: Set2Set (Vinyals et al., 2015). Node-dropping pooling: SortPool (Zhang et al., 2018), SAGPool (Lee et al., 2019), TopKPool (Gao & Ji, 2019), ASAP (Ranjan et al., 2020). Clustering-based pooling: DiffPool (Ying et al., 2018), MinCutPool (Bianchi et al., 2020), HaarPool (Wang et al., 2020), StructPool (Yuan & Ji, 2020). EdgePool (Diehl, 2019) merges nodes along high-scoring edges. Attention-based global pooling: GMT (Baek et al., 2021). Additional models: SOPool (Wang & Ji, 2023), HAP (Liu et al., 2021), PAS (Wei et al., 2021), GMN (Ahmadi, 2020), DKEPool (Chen et al., 2023), JKNet (Xu et al., 2018). On TUdatasets, we also include five kernel baselines: GK (Shervashidze et al., 2009), RW (Vishwanathan et al., 2010), WL subtree (Shervashidze et al., 2011), DGK (Yanardag & Vishwanathan, 2015), and AWE (Ivanov & Burnaev, 2018). An overview of baseline characteristics versus HISTOGRAPH appears in Table 1.

**Benchmarks.** We use the OGB benchmark (Hu et al., 2020) and the widely used TUDatasets (Morris et al., 2020); dataset statistics appear in Tables 7 and 8 in Appendix A. For OGB, we follow Baek et al.

Table 2: Comparison of graph-classification accuracy (%) ↑ on TU datasets with HISTOGRAPH and existing benchmark methods. All methods use a 5-layer GIN backbone. Only top-three methods (plus JKNet) are shown and marked First, Second, Third. Additional results and methods appear in Table 9 in Appendix F.

| Method ↓ / Dataset → | IMDB-B | IMDB-M | MUTAG | PTC | PROTEINS | RDT-B | NCI1 |
|---|---|---|---|---|---|---|---|
| SOPool (Wang & Ji, 2023) | $78.5_{\pm2.8}$ | $54.6_{\pm3.6}$ | $95.3_{\pm4.4}$ | $75.0_{\pm4.3}$ | $80.1_{\pm2.7}$ | $91.7_{\pm2.7}$ | $\mathbf{84.5_{\pm1.3}}$ |
| GMT (Baek et al., 2021) | $\mathbf{79.5_{\pm2.5}}$ | $55.0_{\pm2.8}$ | $95.8_{\pm3.2}$ | $74.5_{\pm4.1}$ | $80.3_{\pm4.3}$ | $93.9_{\pm1.9}$ | $84.1_{\pm2.1}$ |
| HAP (Liu et al., 2021) | $79.1_{\pm2.8}$ | $\mathbf{55.3_{\pm2.6}}$ | $95.2_{\pm2.8}$ | $75.2_{\pm3.6}$ | $79.9_{\pm4.3}$ | $92.2_{\pm2.5}$ | $81.3_{\pm1.8}$ |
| PAS (Wei et al., 2021) | $77.3_{\pm4.1}$ | $53.7_{\pm3.1}$ | $94.3_{\pm5.5}$ | $71.4_{\pm3.9}$ | $78.5_{\pm2.5}$ | $93.7_{\pm1.3}$ | $82.8_{\pm2.2}$ |
| HaarPool (Wang et al., 2020) | $79.3_{\pm3.4}$ | $53.8_{\pm3.0}$ | $90.0_{\pm3.6}$ | $73.1_{\pm5.0}$ | $\mathbf{80.4_{\pm1.8}}$ | $93.6_{\pm1.1}$ | $78.6_{\pm0.5}$ |
| GMN (Ahmadi, 2020) | $76.6_{\pm4.5}$ | $54.2_{\pm2.7}$ | $\mathbf{95.7_{\pm4.0}}$ | $\mathbf{76.3_{\pm4.3}}$ | $79.5_{\pm3.5}$ | $93.5_{\pm0.7}$ | $82.4_{\pm1.9}$ |
| DKEPool (Chen et al., 2023) | $80.9_{\pm2.3}$ | $56.3_{\pm2.0}$ | $97.3_{\pm3.6}$ | $79.6_{\pm4.0}$ | $81.2_{\pm3.8}$ | $95.0_{\pm1.0}$ | $85.4_{\pm2.3}$ |
| JKNet (Xu et al., 2018) | $78.5_{\pm2.0}$ | $54.5_{\pm2.0}$ | $93.0_{\pm3.5}$ | $72.5_{\pm2.0}$ | $78.0_{\pm1.5}$ | $91.5_{\pm2.0}$ | $82.0_{\pm1.5}$ |
| HISTOGRAPH (Ours) | $\mathbf{87.2_{\pm1.7}}$ | $\mathbf{61.9_{\pm5.5}}$ | $\mathbf{97.9_{\pm3.5}}$ | $79.1_{\pm4.8}$ | $\mathbf{97.8_{\pm0.4}}$ | $93.4_{\pm0.9}$ | $\mathbf{85.9_{\pm1.8}}$ |

Table 3: Comparison of graph classification ROC-AUC (%) ↑ on different datasets between HISTOGRAPH and existing baselines on OGB datasets. All methods use a 3-layer GCN backbone for fair comparison. Only the top three methods are included and marked by First, Second, and Third. Additional methods are presented in Table 10 in Appendix F.

| | [†] symbolizes non-learnable methods. | | | |
|---|---|---|---|---|
| Method ↓ / Dataset → | MOLHIV | MOLBBBP | MOLTOX21 | TOXCAST |
| GCN[†] (Kipf & Welling, 2016) | $76.18_{\pm1.26}$ | $65.67_{\pm1.86}$ | $75.04_{\pm0.80}$ | $60.63_{\pm0.51}$ |
| GIN[†] (Xu et al., 2019) | $75.84_{\pm1.35}$ | $66.78_{\pm1.77}$ | $73.27_{\pm0.84}$ | $60.83_{\pm0.46}$ |
| MinCutPool (Bianchi et al., 2020) | $75.37_{\pm2.05}$ | $65.97_{\pm1.13}$ | $75.11_{\pm0.69}$ | $\mathbf{62.48_{\pm1.33}}$ |
| GMT (Baek et al., 2021) | $77.56_{\pm1.25}$ | $\mathbf{68.31_{\pm1.62}}$ | $\mathbf{77.30_{\pm0.59}}$ | $65.44_{\pm0.58}$ |
| HAP (Liu et al., 2021) | $75.71_{\pm1.33}$ | $66.01_{\pm1.43}$ | - | - |
| PAS (Wei et al., 2021) | $\mathbf{77.68_{\pm1.28}}$ | $66.97_{\pm1.21}$ | - | - |
| DKEPool (Chen et al., 2023) | $\mathbf{78.65_{\pm1.19}}$ | $\mathbf{69.73_{\pm1.51}}$ | - | - |
| HISTOGRAPH (Ours) | $\mathbf{77.81_{\pm0.89}}$ | $\mathbf{72.02_{\pm1.46}}$ | $\mathbf{77.49_{\pm0.70}}$ | $\mathbf{66.35_{\pm0.80}}$ |

(2021); Chen et al. (2023) with 3 GCN layers; for TUDatasets, we adopt Wang & Ji (2023); Chen et al. (2023); Gao & Ji (2019); Gao et al. (2021), typically using 5 GIN layers. For deeper variants, we keep the backbone and vary the number of layers. Hyperparameters are in Appendix C.1. Additionally, we benchmark HISTOGRAPH on several node-classification datasets spanning heterophilic and homophilic graphs (Table 11) and across varying GNN depths (Table 4). Further results appear in Appendix F: feature-distance across layers for GCN and GCN with HISTOGRAPH (Table 12), comparison to the GraphGPS baseline (Table 14), and link prediction (Table 13).

## 5.1 END-TO-END ACTIVATION AGGREGATION WITH HISTOGRAPH

We evaluate end-to-end activation aggregation with HISTOGRAPH on graph-level benchmarks and node classification. We first report results on TUDatasets (Table 2), followed by OGB molecular property prediction (Table 3), and finally depth-scaled node classification (Table 4).

**TUDatasets.** On seven datasets (Morris et al., 2020) (IMDB-B, IMDB-M, MUTAG, PTC, PROTEINS, RDT-B, NCI1), HISTOGRAPH attains state-of-the-art performance on 5 of 7: IMDB-B 87.2%, IMDB-M 61.9%, MUTAG 97.9%, PROTEINS 97.8%, NCI1 85.9%. It is marginally behind on PTC at 79.1% versus 79.6% for DKEPool. Relative to the second-best method, gains are substantial on PROTEINS (+16.6%), IMDB-B (+6.3%), and IMDB-M (+5.6%). Although DKEPool slightly leads on PTC and RDT-B, the overall trend favors HISTOGRAPH across diverse graph classification benchmarks.

The large gain on PROTEINS (+16.6%) is driven by all three HISTOGRAPH components: the ablation in Table 6 shows that removing any one of signed normalization, layer-wise, or node-wise attention substantially degrades performance, and Table 16 confirms that uniform averaging and randomized attention fall far short. We provide a detailed discussion in Appendix I.

Table 4: Node classification accuracy (%) on benchmark datasets with varying GNN depth.

| Dataset | Method | 2 | 4 | 8 | 16 | 32 | 64 |
|---------|--------|---|---|---|----|----|----|
| Cora | GCN | 81.1 | 80.4 | 69.5 | 64.9 | 60.3 | 28.7 |
| | GCN + HISTOGRAPH | **81.3** | **82.9** | **80.7** | **83.1** | **80.6** | **77.5** |
| Citeseer | GCN | 70.8 | 67.6 | 30.2 | 18.3 | 25.0 | 20.0 |
| | GCN + HISTOGRAPH | **70.9** | **69.5** | **69.9** | **69.3** | **67.2** | **63.4** |
| Pubmed | GCN | **79.0** | 76.5 | 61.2 | 40.9 | 22.4 | 35.3 |
| | GCN + HISTOGRAPH | 78.9 | **78.2** | **78.6** | **80.4** | **80.0** | **79.3** |

**OGB molecular property prediction.** On four OGB datasets (Hu et al., 2020) (MOLHIV, MOLTOX21, TOXCAST, MOLBBBP), HISTOGRAPH achieves the top ROC-AUC on 3 of 4: MOLBBBP 72.02%, MOLTOX21 77.49%, TOXCAST 66.35%. Margins over the second-best are +2.29% on MOLBBBP versus DKEPool, +0.91% on TOXCAST versus GMT, and +0.19% on MOLTOX21 versus GMT. On MOLHIV, DKEPool leads with 78.65%, while HISTOGRAPH is competitive at 77.81%, ranking in the top three, indicating strong generalization across molecular property prediction.

**Node classification.** Table 4 shows that HISTOGRAPH mitigates over-smoothing: standard GCN accuracy degrades with depth, whereas HISTO-GRAPH maintains stable, competitive performance up to 64 layers. This improves feature propagation while preserving discriminative power, particularly on heterophilic graphs. Additional node-classification results for heterophilic and homophilic datasets appear in Table 11 in Appendix F.

Table 5: Graph classification accuracy (%) ↑ summary. More results are reported in Table 17.

| Dataset | Method | Acc. |
|---------|--------|------|
| IMDB-M | MeanPool | 54.7 |
| | FT | 67.3 |
| | Full FT | 62.7 |
| | End-to-End | 61.9 |
| IMDB-B | MeanPool | 76.0 |
| | FT | 94.0 |
| | Full FT | 94.0 |
| | End-to-End | 87.2 |
| PROTEINS | MeanPool | 75.9 |
| | FT | 97.3 |
| | Full FT | 97.3 |
| | End-to-End | 97.8 |
| PTC | MeanPool | 77.1 |
| | FT | 85.7 |
| | Full FT | 97.1 |
| | End-to-End | 88.6 |

## 5.2 POST-PROCESSING OF TRAINED GNNS WITH HISTOGRAPH

We evaluate HISTOGRAPH as a lightweight post-processing strategy on four TU graph-classification datasets: IMDB-B, IMDB-M, PROTEINS, and PTC. For each dataset, we train GINs with 5, 16, 32, and 64 layers using standard architectures and mean pooling. After convergence, we save per-fold checkpoints and apply HISTOGRAPH in three modes: (i) auxiliary head on a frozen backbone (HISTOGRAPH(FT)), (ii) full joint fine-tuning (HISTOGRAPH(Full FT)), and (iii) end-to-end training from scratch for comparison. Complete depth-wise results appear in Table 17 in Appendix F.

Table 5 summarizes the graph-classification accuracy (%) across GIN depths for each dataset and method. HISTOGRAPH used as a frozen auxiliary head (FT) consistently improves performance vs. MeanPool, often matching or surpassing full fine-tuning (Full FT) and end-to-end training. For example, on IMDB-M, FT raises accuracy from 54.7% (MeanPool) to 67.3%; on IMDB-B, both FT and Full FT reach 94.0%, far above the baseline (76.0%) and end-to-end (87.2%). On PROTEINS, all HISTOGRAPH variants achieve near-optimal performance, demonstrating effectiveness across datasets of varying size and characteristics. On PTC, Full FT attains the best score (97.1%), showing joint fine-tuning can further enhance results. Overall, HISTOGRAPH offers a flexible, effective post-processing strategy that consistently boosts GNN performance.

**Runtime Analysis.** We measure average training and inference time per epoch for GCN backbones with 3 and 32 layers on MOLHIV and TOXCAST, comparing MeanPool, End-to-End, and FT. As shown in Fig. 4, End-to-End is costlier than MeanPool during training (e.g., 60.34s vs. 41.27s for 32 layers on MOLHIV) yet remains scalable. FT, which fine-tunes only the head on a pretrained MeanPool model, cuts overhead: training time is slightly higher for 3 layers but significantly lower for 32 layers on both datasets. At inference time, HISTOGRAPH adds negligible overhead over MeanPool since no backward pass is required and the forward cost of the HISTOGRAPH head ($O(NLD + N^2D)$) is small relative to the GNN backbone on the molecular-scale graphs evaluated (see Table 18 in Appendix J for detailed inference times). While achieving results comparable to End-to-End (Table 17), FT offers an efficient way to boost existing models. Finally, HISTOGRAPH is significantly faster than GMT (Baek et al., 2021) in almost all cases, with larger speedups for deeper networks.

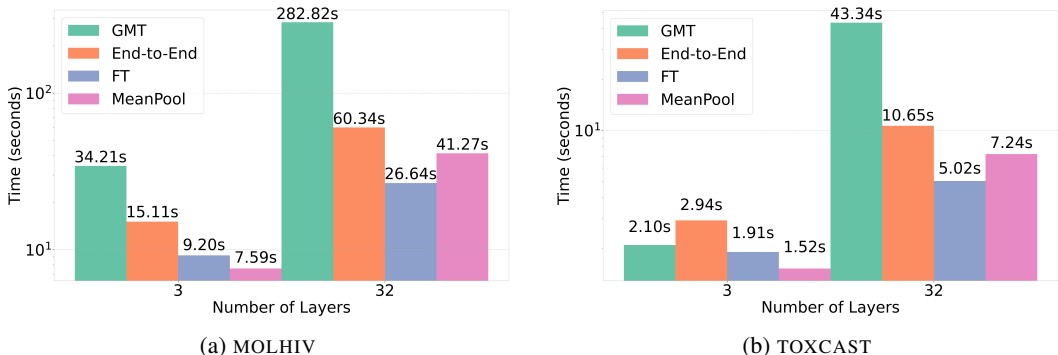

(a) MOLHIV                                      (b) TOXCAST

Figure 4: Average training time per epoch (in log scale) for GCN backbones with 3 and 32 layers, evaluated on the MOLHIV and TOXCAST datasets. Each configuration is compared across four post-processing methods: GMT(Baek et al., 2021), MeanPool, HISTOGRAPH, and HISTOGRAPH-FT.

## 5.3 ABLATION STUDY

**Setup.** We assess component contributions on the TUDatasets PROTEINS dataset by removing or modifying parts and measuring classification accuracy (Table 6). We test three variants: (i) removing division-by-sum normalization, (ii) disabling layer-wise attention that models inter-layer dependencies, and (iii) disabling node-wise attention that captures cross-node dependencies.

**Results and discussion.** On PROTEINS, HISTO-GRAPH attains 97.80% accuracy with a 0.40 standard deviation. Every ablation reduces accuracy; remov-

Table 6: Ablation on the PROTEINS dataset. Each row shows the performance of a HISTO-GRAPH variant with a component removed.

| Variant | Acc. (%) | Std. |
|---|---|---|
| DKEPool | 81.20 | 3.80 |
| w/o Division by Sum | 74.45 | 6.28 |
| w/o Layer-wise Attention | 78.61 | 4.82 |
| w/o Node-wise Attention | 80.78 | 7.71 |
| HISTOGRAPH (Ours) | 97.80 | 0.40 |

ing division-by-sum normalization performs worst at 74.45% ± 6.28, indicating each component is necessary. Removing layer-wise normalization allows attention weights to grow unbounded, destabilizing training and overshadowing early discriminative layers. Our signed normalization balances layer contributions and enables additive and subtractive filtering (Section 4), preserving discriminative information and stability. Against alternative aggregation strategies (mean aggregation and randomized attention), HISTOGRAPH consistently outperforms them by a significant margin (Table 16, Appendix F). Overall, normalization, layer-wise attention, and node-wise attention are critical for capturing complex dependencies and realizing the full performance of HISTOGRAPH.

## 6 CONCLUSION

We introduced HISTOGRAPH, a two-stage attention-based pooling layer that learns from historical activations to produce stronger graph-level representations. The design is simple and principled: layer-wise attention captures the evolution of each node's trajectory across depths, node-wise self-attention models spatial interactions at readout, and signed layer-wise normalization balances contributions across layers to preserve discriminative signals and stabilize training. This combination mitigates over-smoothing and supports deeper GNNs while keeping computation and memory overhead modest. Across TU and OGB graph-level benchmarks, node-classification settings, and link prediction (OGBL-COLLAB), HISTOGRAPH consistently improves over strong pooling baselines and matches or surpasses leading methods on multiple datasets. Moreover, used as a lightweight post-processing head on frozen backbones, HISTOGRAPH delivers additional gains without retraining the encoder. The current design is best suited for small-to-medium-sized graphs; extending HISTOGRAPH to very large graphs via sparse attention or hierarchical coarsening is a promising direction for future work. Taken together, the results establish intermediate activations as a valuable signal for readout and position HISTOGRAPH as a practical, general drop-in pooling layer for modern GNNs.

**Acknowledgements**   HM is supported by the Israel Science Foundation through a personal grant (ISF 264/23) and an equipment grant (ISF 532/23), and by the Career Advancement Chairs in Artificial Intelligence – Schmidt Futures. ME acknowledges support from the Israeli Ministry of Innovation, Science & Technology.

**Reproducibility Statement**   To ensure reproducibility, we provide all code, model architectures, training scripts, and hyperparameter settings in a public repository (available upon acceptance). Dataset preprocessing, splits, and downsampling are detailed in Appendix A. Hyperparameter configurations, including batch sizes, learning rates, hidden dimensions, and model depths, are documented in Appendix C.1. Experiments were conducted using PyTorch and PyTorch Geometric on NVIDIA L40, A100, and GeForce RTX 4090 GPUs, with Weights and Biases for logging and model selection. All random seeds and training protocols are specified to facilitate replication.

**Ethics Statement**   Our work involves minimal ethical concerns. We use publicly available datasets (TU, OGB) that are widely adopted in graph learning research, adhering to their licensing terms. No private or sensitive data is introduced. Our method is primarily methodological, but we encourage responsible use to avoid potential misuse in applications that could impact privacy or enable harm. We acknowledge the environmental impact of large-scale training and note that HISTOGRAPH's computational efficiency may reduce energy costs compared to retraining full models.

**Usage of Large Language Models in This Work**   Large language models were used solely for minor text editing suggestions to improve clarity and grammar. All research concepts, code development, experimental design, and original writing were performed by the authors.

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

## A  DATASET STATISTICS

Tables 7 and 8 summarize the statistics of the datasets used in our experiments. Table 7 covers molecular property prediction datasets from the Open Graph Benchmark (OGB), including MOLHIV, MOLBBBP, MOLTOX21, and TOXCAST, reporting the number of graphs, number of prediction classes, and average number of nodes per graph. Table 8 presents the statistics of graph classification datasets from the TU benchmark suite, including social network datasets (IMDB-B, IMDB-M) and bioinformatics datasets (MUTAG, PTC, PROTEINS, RDT-B, NCI1). These datasets vary widely in graph sizes and label space, providing a comprehensive evaluation setting across small, medium, and large graphs with diverse class distributions.

Table 7: Dataset statistics: number of graphs, number of classes, and average number of nodes.

| Dataset | MOLHIV | MOLBBBP | MOLTOX21 | TOXCAST |
|---|---|---|---|---|
| # Graphs | 41,127 | 2,039 | 7,831 | 8,576 |
| # Classes | 2 | 2 | 12 | 617 |
| Nodes (avg.) | 25.51 | 24.06 | 18.57 | 18.78 |

Table 8: Statistics of TU benchmark datasets.

| | | IMDB-B | IMDB-M | MUTAG | PTC | PROTEINS | RDT-B | NCI1 |
|---|---|---|---|---|---|---|---|---|
| **Dataset** | # Graphs | 1000 | 1500 | 188 | 344 | 1113 | 2000 | 4110 |
| | # Classes | 2 | 3 | 2 | 2 | 2 | 2 | 2 |
| | Nodes(max) | 136 | 89 | 28 | 109 | 620 | 3783 | 111 |
| | Nodes(avg.) | 19.8 | 13.0 | 18.0 | 25.6 | 39.1 | 429.6 | 29.2 |

## B  IMPLEMENTATION DETAILS OF HISTOGRAPH

Algorithm 1 outlines the forward pass of HISTOGRAPH. The input $\mathbf{X} \in \mathbb{R}^{N \times L \times D_{\text{in}}}$ consists of node embeddings across $L$ GNN layers, for $N$ nodes per graph (referred to as historical graph activations). We first project the input to a common hidden dimension $D$ using a shared linear transformation. Sinusoidal positional encodings are added to encode the layer index. The final-layer embeddings serve as the query in an attention mechanism, while all intermediate layers act as key and value inputs. Attention scores are computed, averaged across nodes, and normalized over layers to yield a weighted aggregation of layer-wise features. A multi-head self-attention (MHSA) block is then applied over the aggregated node representations to capture spatial dependencies. Finally, a global average pooling operation over the node dimension produces the final graph-level representation $\mathbf{Y} \in \mathbb{R}^{D}$.

To stabilize training, we combined the output of HISTOGRAPH with a simple mean pooling baseline using a learnable weighting factor $\alpha \in [0, 1]$. Specifically, the final graph representation was computed as a convex combination of the output of our method and the mean of the final-layer node embeddings: $\mathbf{Y}_{\text{final}} = \alpha \cdot \mathbf{Y}_{\text{HISTOGRAPH}} + (1 - \alpha) \cdot \mathbf{Y}_{\text{mean}}$. We experimented both with fixed and learnable values of $\alpha$, and found that incorporating the mean-pooling signal helps guide the optimization in early training stages.

## C  EXPERIMENTAL DETAILS

We implemented our method using PyTorch (Paszke, 2019) (offered under BSD-3 Clause license) and the PyTorch Geometric library (Fey & Lenssen, 2019) (offered under MIT license). All experiments were run on NVIDIA L40, NVIDIA A100 and GeForce RTX 4090 GPUs. For logging, hyperparameter tuning, and model selection, we used the Weights and Biases (W&B) framework (Biewald, 2020).

In the subsection below, we provide details on the hyperparameter configurations used across our experiments.

---

**Algorithm 1** HISTOGRAPH Forward Pass

---

**Input:** $\mathbf{X} \in \mathbb{R}^{N \times L \times D_{\text{in}}}$
**Output:** Graph-level representation $\mathbf{Y} \in \mathbb{R}^D$
$\mathbf{X}' \leftarrow \text{Emb}_{\text{hist}}(\mathbf{X})$      ▷ Linear projection to $D$ dimensions
$\widetilde{\mathbf{X}} \leftarrow \mathbf{X}' + \mathbf{P}$      ▷ Add sinusoidal positional encoding
$\mathbf{Q} \leftarrow W^Q \widetilde{\mathbf{X}}_{L-1}$      ▷ Query: last-layer embedding
$\mathbf{K} \leftarrow W^K \widetilde{\mathbf{X}}, \quad \mathbf{V} \leftarrow \widetilde{\mathbf{X}}$      ▷ Key and Value: all layers
$\mathbf{c} \leftarrow \frac{\mathbf{Q}\mathbf{K}^\top}{\sqrt{D}}$      ▷ Dot-product attention logits
$\mathbf{c} \leftarrow \text{Average}(\mathbf{c})$      ▷ Average across nodes
$\alpha_t \leftarrow \frac{c_t}{\sum_{t'} c_{t'}}$      ▷ Normalize over time
$\mathbf{H} \leftarrow \sum_{t=0}^{L-1} \alpha_t \widetilde{\mathbf{X}}_t$      ▷ Layer-wise aggregation
$\mathbf{Z} \leftarrow \text{MHSA}(\mathbf{H}, \mathbf{H}, \mathbf{H})$      ▷ Node-wise self-attention
**return** $\mathbf{Y} = \text{Average}(\mathbf{Z})$      ▷ Average across nodes

---

## C.1 HYPERPARAMETERS

The hyperparameters in our method include the batch size $B$, hidden dimension $D$, learning rate $l$, and weight decay $\gamma$. We also tune architectural and attention-specific components such as the number of attention heads $H$, use of fully connected layers, inclusion of zero attention token, use of layer normalization, and skip connections. Attention dropout rates are controlled via the multi-head attention dropout $p_{\text{mha}}$ and attention mask dropout $p_{\text{mask}}$. We further include the use of a learning rate schedule as a hyperparameter. Additionally, we consider different formulations for the attention coefficient parameterization $\alpha_{\text{type}}$, including learnable, fixed, and gradient-constrained variants. Hyperparameters were selected via a combination of grid search and Bayesian optimization, using validation performance as the selection criterion. For baseline models, we consider the search space of their relevant hyperparameters.

## D ADDITIONAL PROPERTIES OF HISTOGRAPH

**Contribution of Node-wise Attention for Graph-Level Prediction.** Let $\mathbf{H} = [\mathbf{h}_1, \ldots, \mathbf{h}_N] \in \mathbb{R}^{N \times D}$ be the cross-layer-pooled node embeddings. Suppose the downstream task requires a function $f : \mathbb{R}^{N \times D} \to \mathbb{R}^K$ that is permutation-invariant but non-uniform (e.g., depends on inter-node interactions). Then, standard mean pooling cannot approximate $f$ unless it includes additional inter-node operations like node-wise attention.

As a concrete example where the node-wise attention is beneficial in our HISTOGRAPH, let us consider a graph $G = (V, E)$ composed of two subgraphs connected by a narrow bridge. Let $G_L = (V_L, E_L)$ be a large graph $G(n, p)$ with $n \gg 1$, and let $G_R = (V_R, E_R)$ be a singleton graph containing a single node $v_R$. The resulting structure is illustrated in Figure 5.

Suppose the graph-level classification task depends solely on the features of the singleton node $v_R$ (e.g., label is determined by a property encoded in $v_R$). In this setting, a naive mean pooling aggregates all node embeddings uniformly. As $n$ increases, the contribution of $v_R$ to the pooled representation becomes increasingly marginal, leading to its signal being dominated by the embeddings from the much larger subgraph $G_L$. This becomes especially problematic when there is a distribution shift at test time, e.g., $G_L$ becomes larger or denser, which further suppresses the contribution of $v_R$.

In contrast, a node-wise attention mechanism can learn to attend selectively to $v_R$, regardless of the size of $G_L$, making it robust to distributional changes. This demonstrates the contribution of node-wise attention in capturing non-uniform importance of nodes in our HISTOGRAPH.

## E PROOFS

### E.1 HISTOGRAPH MITIGATES OVERSMOOTHING

**Proof.**

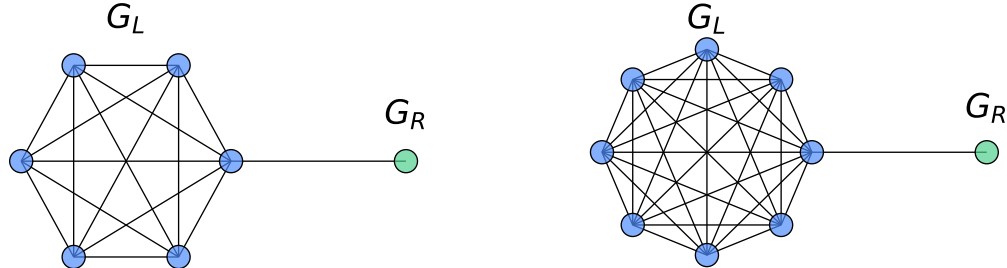

Figure 5: Barbell graph illustrating a distribution shift: a singleton node (right) is connected to a larger subgraph (left) whose size increases at test time (blue) compared to training (green). Node-wise attention helps preserve the importance of the singleton node despite the dominance of the larger subgraph.

Consider two nodes $u$ and $v$. Under the definition of HISTOGRAPH's final embedding:

$$h_u - h_v = \sum_{l=0}^{L-1} \alpha_l \left( \mathbf{x}_u^{(l)} - \mathbf{x}_v^{(l)} \right). \tag{13}$$

We split the sum into layers before and after $L_0$:

$$h_u - h_v = \sum_{l=0}^{L_0} \alpha_l \left( \mathbf{x}_u^{(l)} - \mathbf{x}_v^{(l)} \right) + \sum_{l=L_0+1}^{L-1} \alpha_l \left( \mathbf{x}_u^{(l)} - \mathbf{x}_v^{(l)} \right). \tag{14}$$

By over-smoothing (Eq. 9), for all $l > L_0$:

$$\|\mathbf{x}_u^{(l)} - \mathbf{x}_v^{(l)}\| \approx 0, \tag{15}$$

and hence the second sum is negligible. Therefore,

$$h_u - h_v \approx \sum_{l=0}^{L_0} \alpha_l \left( \mathbf{x}_u^{(l)} - \mathbf{x}_v^{(l)} \right). \tag{16}$$

Because initial node representations differ (a standard assumption for distinct nodes), there exists at least one layer $l' \leq L_0$ for which

$$\|\mathbf{x}_u^{(l')} - \mathbf{x}_v^{(l')}\| \neq 0. \tag{17}$$

Given that HISTOGRAPH employs learned dynamic attention, suppose $\alpha_{l'} \neq 0$. Consequently:

$$\|h_u - h_v\| \approx \left\| \alpha_{l'} \left( \mathbf{x}_u^{(l')} - \mathbf{x}_v^{(l')} \right) + \sum_{l \neq l'} \alpha_l \left( \mathbf{x}_u^{(l)} - \mathbf{x}_v^{(l)} \right) \right\| \tag{18}$$

$$> 0. \tag{19}$$

This directly contradicts the assumption that node embeddings become indistinguishable in the pooled representation. Thus, HISTOGRAPH mitigates over-smoothing by explicitly retaining discriminative early-layer representations.

## F ADDITIONAL EXPERIMENTS

Table 9 and Table 10 show the results of different methods on two different settings of multiple graph pooling methods. Table 11 reports node classification accuracy on both heterophilic and homophilic datasets. We observe that our method (HISTOGRAPH+ GCN) consistently outperforms standard GCN and JKNet across all datasets. The improvements are particularly pronounced on heterophilic graphs such as Actor, Squirrel, and Chameleon, where our method achieves gains of up to 12.3% over

Table 9: Comparison of graph classification accuracy (%) ↑ on different datasets with HISTO-GRAPH and existing benchmark graph classification methods on TU datasets. All methods use a 5-layer GIN backbone for fair comparison. Top three results are marked as First, Second, and **Third**.

| Method ↓ / Dataset → | IMDB-B | IMDB-M | MUTAG | PTC | PROTEINS | RDT-B | NCI1 |
|---|---|---|---|---|---|---|---|
| **KERNEL** | | | | | | | |
| GK (Shervashidze et al., 2009) | $65.9_{\pm1.0}$ | $43.9_{\pm0.4}$ | $81.4_{\pm1.7}$ | $57.3_{\pm1.4}$ | $71.7_{\pm0.6}$ | $77.3_{\pm0.2}$ | $62.3_{\pm0.3}$ |
| RW (Vishwanathan et al., 2010) | - | - | $79.2_{\pm2.1}$ | $57.9_{\pm1.3}$ | $74.2_{\pm0.4}$ | - | - |
| WL (Shervashidze et al., 2011) | $73.8_{\pm3.9}$ | $50.9_{\pm3.8}$ | $82.1_{\pm0.4}$ | $60.0_{\pm0.5}$ | $74.7_{\pm0.5}$ | - | $82.2_{\pm0.2}$ |
| DGK (Yanardag & Vishwanathan, 2015) | $67.0_{\pm0.6}$ | $44.6_{\pm0.5}$ | - | $60.1_{\pm2.6}$ | $75.7_{\pm0.5}$ | $78.0_{\pm0.4}$ | $80.3_{\pm0.5}$ |
| AWE (Ivanov & Burnaev, 2018) | $74.5_{\pm5.9}$ | $51.5_{\pm3.6}$ | $87.9_{\pm9.8}$ | - | - | $87.9_{\pm2.5}$ | - |
| **GNN** | | | | | | | |
| ASAP (Ranjan et al., 2020) | $77.6_{\pm2.1}$ | $54.5_{\pm2.1}$ | $91.6_{\pm5.3}$ | $72.4_{\pm7.5}$ | $78.3_{\pm4.0}$ | $93.1_{\pm2.1}$ | $75.1_{\pm1.5}$ |
| SOPool (Wang & Ji, 2023) | $78.5_{\pm2.8}$ | $54.6_{\pm3.6}$ | $95.3_{\pm4.4}$ | $75.0_{\pm4.3}$ | $80.1_{\pm2.7}$ | $91.7_{\pm2.7}$ | **$84.5_{\pm1.3}$** |
| GMT (Baek et al., 2021) | **$79.5_{\pm2.5}$** | $55.0_{\pm2.8}$ | $95.8_{\pm3.2}$ | $74.5_{\pm4.1}$ | $80.3_{\pm4.3}$ | $93.9_{\pm1.9}$ | $84.1_{\pm2.1}$ |
| HAP (Liu et al., 2021) | $79.1_{\pm2.8}$ | **$55.3_{\pm2.6}$** | $95.2_{\pm2.8}$ | $75.2_{\pm3.6}$ | $79.9_{\pm4.3}$ | $92.2_{\pm2.5}$ | $81.3_{\pm1.8}$ |
| PAS (Wei et al., 2021) | $77.3_{\pm4.1}$ | $53.7_{\pm3.1}$ | $94.3_{\pm5.5}$ | $71.4_{\pm3.9}$ | $78.5_{\pm2.5}$ | $93.7_{\pm1.3}$ | $82.8_{\pm2.2}$ |
| HaarPool (Wang et al., 2020) | $79.3_{\pm3.4}$ | $53.8_{\pm3.0}$ | $90.0_{\pm3.6}$ | $73.1_{\pm5.0}$ | $80.4_{\pm1.8}$ | $93.6_{\pm1.1}$ | $78.6_{\pm0.5}$ |
| DiffPool (Ying et al., 2018) | $73.9_{\pm3.6}$ | $50.7_{\pm2.9}$ | $94.8_{\pm4.8}$ | $68.3_{\pm5.9}$ | $76.2_{\pm3.1}$ | $91.8_{\pm2.1}$ | $76.6_{\pm1.3}$ |
| GMN (Ahmadi, 2020) | $76.6_{\pm4.5}$ | $54.2_{\pm2.7}$ | $95.7_{\pm4.0}$ | **$76.3_{\pm4.3}$** | $79.5_{\pm3.5}$ | $93.5_{\pm0.7}$ | $82.4_{\pm1.9}$ |
| DKEPool (Chen et al., 2023) | $80.9_{\pm2.3}$ | $56.3_{\pm2.0}$ | $97.3_{\pm3.6}$ | $79.6_{\pm4.0}$ | $81.2_{\pm3.8}$ | $95.0_{\pm1.0}$ | $85.4_{\pm2.3}$ |
| JKNet (Xu et al., 2018) | $78.5_{\pm2.0}$ | $54.5_{\pm2.0}$ | $93.0_{\pm3.5}$ | $72.5_{\pm2.0}$ | $78.0_{\pm1.5}$ | $91.5_{\pm2.0}$ | $82.0_{\pm1.5}$ |
| HISTOGRAPH (Ours) | $87.2_{\pm1.7}$ | $61.9_{\pm5.5}$ | $97.9_{\pm3.5}$ | $79.1_{\pm4.8}$ | $97.8_{\pm0.4}$ | $93.4_{\pm0.9}$ | $85.9_{\pm1.8}$ |

Table 10: Comparison of graph classification ROC-AUC (%) ↑ on different datasets between HISTO-GRAPH and existing baselines on OGB datasets. All methods use a 3-layer GCN backbone for fair comparison. The metric used is ROC-AUC. The top three methods are marked by First, Second, and **Third**.

| Method ↓ / Dataset → | MOLHIV | MOLBBBP | MOLTOX21 | TOXCAST |
|---|---|---|---|---|
| [†] symbolizes non-learnable methods. | | | | |
| GCN[†] (Kipf & Welling, 2016) | $76.18_{\pm1.26}$ | $65.67_{\pm1.86}$ | $75.04_{\pm0.80}$ | $60.63_{\pm0.51}$ |
| GIN[†] (Xu et al., 2019) | $75.84_{\pm1.35}$ | $66.78_{\pm1.77}$ | $73.27_{\pm0.84}$ | $60.83_{\pm0.46}$ |
| HaarPool[†] (Wang et al., 2020) | $74.69_{\pm1.62}$ | $66.11_{\pm0.82}$ | - | - |
| ASAP (Ranjan et al., 2020) | $72.86_{\pm1.40}$ | $63.50_{\pm2.47}$ | $72.24_{\pm1.66}$ | $58.09_{\pm1.62}$ |
| TopKPool (Gao & Ji, 2019) | $72.27_{\pm0.91}$ | $65.19_{\pm2.30}$ | $69.39_{\pm2.02}$ | $58.42_{\pm0.91}$ |
| SortPool (Zhang et al., 2018) | $71.82_{\pm1.63}$ | $65.98_{\pm1.70}$ | $69.54_{\pm0.75}$ | $58.69_{\pm1.71}$ |
| JKNet (Xu et al., 2018) | $74.99_{\pm1.60}$ | $65.62_{\pm0.77}$ | $65.98_{\pm0.46}$ | - |
| SAGPool (Lee et al., 2019) | $74.56_{\pm1.69}$ | $65.16_{\pm1.93}$ | $71.10_{\pm1.06}$ | $59.88_{\pm0.79}$ |
| Set2Set (Vinyals et al., 2015) | $74.70_{\pm1.65}$ | $66.79_{\pm1.05}$ | $74.10_{\pm1.13}$ | $59.70_{\pm1.04}$ |
| SAGPool(H) (Lee et al., 2019) | $71.44_{\pm1.67}$ | $63.94_{\pm2.59}$ | $69.81_{\pm1.75}$ | $58.91_{\pm0.80}$ |
| EdgePool (Diehl, 2019) | $72.66_{\pm1.70}$ | $67.18_{\pm1.97}$ | $73.77_{\pm0.68}$ | $60.70_{\pm0.92}$ |
| MinCutPool (Bianchi et al., 2020) | $75.37_{\pm2.05}$ | $65.97_{\pm1.13}$ | $75.11_{\pm0.69}$ | **$62.48_{\pm1.33}$** |
| StructPool (Yuan & Ji, 2020) | $75.85_{\pm1.81}$ | $67.01_{\pm2.65}$ | $75.43_{\pm0.79}$ | $62.17_{\pm1.61}$ |
| SOPool (Wang & Ji, 2023) | $76.98_{\pm1.11}$ | $65.82_{\pm1.66}$ | - | - |
| GMT (Baek et al., 2021) | $77.56_{\pm1.25}$ | **$68.31_{\pm1.62}$** | $77.30_{\pm0.59}$ | $65.44_{\pm0.58}$ |
| HAP (Liu et al., 2021) | $75.71_{\pm1.33}$ | $66.01_{\pm1.43}$ | - | - |
| PAS (Wei et al., 2021) | **$77.68_{\pm1.28}$** | $66.97_{\pm1.21}$ | - | - |
| DiffPool (Ying et al., 2018) | $75.64_{\pm1.86}$ | $68.25_{\pm0.96}$ | $74.88_{\pm0.81}$ | $62.28_{\pm0.56}$ |
| GMN (Ahmadi, 2020) | $77.25_{\pm1.70}$ | $67.06_{\pm1.05}$ | - | - |
| DKEPool (Chen et al., 2023) | $78.65_{\pm1.19}$ | $69.73_{\pm1.51}$ | - | - |
| HISTOGRAPH (Ours) | $77.81_{\pm0.89}$ | $72.02_{\pm1.46}$ | $77.49_{\pm0.70}$ | $66.35_{\pm0.80}$ |

GCN. On homophilic datasets like Cora, Citeseer, and Pubmed, we also observe consistent, albeit smaller, improvements.

To evaluate the ability of HISTOGRAPH to mitigate oversmoothing, we measure the feature distance across layers for a standard GCN, both with and without HISTOGRAPH. The results, presented in

Table 11: Node classification accuracy (mean $\pm$ std) on heterophilic and homophilic datasets.

| Dataset | GCN | JKNet | HISTOGRAPH+ GCN (Ours) |
|---------|-----|-------|------------------------|
| Actor | $27.3 \pm 1.1$ | $35.1 \pm 1.4$ | $\mathbf{36.2 \pm 1.1}$ |
| Squirrel | $53.4 \pm 2.0$ | $45.0 \pm 1.7$ | $\mathbf{65.7 \pm 1.4}$ |
| Chameleon | $64.8 \pm 2.2$ | $63.8 \pm 2.3$ | $\mathbf{69.8 \pm 1.8}$ |
| Cora | $81.1 \pm 0.8$ | $81.0 \pm 1.0$ | $\mathbf{83.1 \pm 0.4}$ |
| Citeseer | $70.8 \pm 0.7$ | $69.8 \pm 0.8$ | $\mathbf{70.9 \pm 0.5}$ |
| Pubmed | $79.0 \pm 0.6$ | $78.1 \pm 0.5$ | $\mathbf{80.4 \pm 0.4}$ |

Table 12: Feature distance metrics across layers, showing the ability of HISTOGRAPH to mitigate oversmoothing. Compared to standard GCN, HISTOGRAPH yields more diverse node embeddings.

| Layer | 0 | 8 | 64 | Final (pre-classifier) |
|-------|---|---|----|------------------------|
| GCN | 2.634 | 2.385 | 1.703 | 1.703 |
| GCN + HISTOGRAPH | 3.710 | 3.403 | 2.888 | 5.000 |

Table 12, show that incorporating HISTOGRAPH consistently leads to higher feature distances across all layers, with the most pronounced improvement observed in the pre-classifier layer as expected due to the oversmoothing.

The results in Tables 13–15 further demonstrate the versatility and effectiveness of HISTO-GRAPH across different tasks and architectures. On the OGBL-COLLAB link prediction benchmark (Table 13), incorporating HISTOGRAPH as a readout function leads to consistent improvements over a standard GCN baseline. Similarly, in molecular property prediction tasks with GraphGPS backbones (Table 14), HISTOGRAPH achieves substantial performance gains across multiple datasets, highlighting its ability to preserve and leverage historical information across layers. Finally, the ablation on the number of historical layers (Table 15) shows that incorporating deeper historical context enhances predictive performance, with the best results obtained when more layers are retained. These findings underscore the robustness of HISTOGRAPH as a drop-in replacement for readout functions across diverse settings. Finally, we present an additional ablation in Table 16, which examines the effect of different aggregation strategies across all layers—mean aggregation, randomized attention, and HISTOGRAPH. Across datasets, HISTOGRAPH consistently achieves superior performance.

### F.1 SCALABLE POST-PROCESSING WITH HISTOGRAPH

Table 17 indicates that all HISTOGRAPH variants consistently outperform the MeanPool baseline for every depth and dataset. In particular, FT, often matches or even exceeds the accuracy of full end-to-end tuning despite having far fewer trainable parameters. For example, at 5 layers it boosts IMDB-M from 54.0 % to 67.3 %, IMDB-B accuracy from 76.0 % to 94.0 %, PROTEINS from 75.0 % to 97.3 %, and PTC from 77.1 % to 85.7 %. As model depth grows, FT remains highly competitive: at 16 layers it achieves 64.7%.

We would like to note that while HISTOGRAPH mitigates over-smoothing by dynamically leveraging early-layer discriminative features, at extreme depths (e.g., 64 layers) we face known optimization challenges in GNNs (Li et al., 2019; Chen et al., 2020; Arroyo et al., 2025). Nonetheless, HISTOGRAPH consistently outperforms baseline pooling methods, as shown in our depth-varying experiments on Cora, Citeseer, and Pubmed in Table 4, demonstrating robustness to model depth.

These findings demonstrate that caching intermediate representations and training a small auxiliary head enables scalable, modular adaptation of GNNs, obtaining strong performance across depths and domains without incurring the computational costs of full model training.

## G  SCALABILITY CONSIDERATIONS

The current design of HISTOGRAPH is best suited for small-to-medium-sized graphs, where the $O(N^2 D)$ node-wise attention and $O(NLD)$ activation storage remain tractable. Our benchmarks

Table 13: Link prediction results on the OGBL-COLLAB dataset. HISTOGRAPH is applied as a drop-in replacement for the readout function with a GCN backbone, demonstrating consistent improvements over the baseline.

| Model | Hits@50 (Test) ↑ | Hits@50 (Validation) ↑ |
|---|---|---|
| Baseline (GCN) | $0.4475 \pm 0.0107$ | $0.5263 \pm 0.0115$ |
| GCN + HISTOGRAPH | $\mathbf{0.4533 \pm 0.0096}$ | $\mathbf{0.5314 \pm 0.0103}$ |

Table 14: Performance comparison of GraphGPS baselines with and without HISTOGRAPH on multiple datasets. Integrating HISTOGRAPH consistently improves performance by preserving layer-wise historical context and enabling adaptive readout.

| Method | PROTEINS | tox21 | ToxCast |
|---|---|---|---|
| GPS + MeanPool | $79.8 \pm 2.1$ | $75.7 \pm 0.4$ | $62.5 \pm 1.09$ |
| GPS + HISTOGRAPH | $\mathbf{98.9 \pm 1.2}$ | $\mathbf{77.8 \pm 2.2}$ | $\mathbf{66.9 \pm 0.69}$ |

span molecular graphs (avg. ∼20 nodes on OGB), social network graphs (avg. ∼20–430 nodes on TUDatasets), citation networks (Cora, Citeseer, Pubmed), and the large-scale OGBL-COLLAB link prediction benchmark. For graphs with significantly more nodes, the quadratic node-wise attention and activation caching become bottlenecks. Several strategies can extend HISTOGRAPH to larger scales: (i) *neighbor sampling* (Hamilton et al., 2017) to restrict the node set per mini-batch, (ii) *sparse attention* mechanisms (Zaheer et al., 2020) that replace the dense $N \times N$ attention with $O(N\sqrt{N})$ or $O(N \log N)$ approximations, and (iii) *hierarchical coarsening* to reduce $N$ before applying node-wise attention. We leave the integration of these techniques with HISTOGRAPH to future work.

## H    INTERPRETABILITY OF LEARNED ATTENTION WEIGHTS

Figure 2 (left) reveals several interpretable patterns in the learned layer-wise attention weights. Across training regimes, the model allocates substantial weight to early layers (particularly layers 0–5), where node embeddings are most discriminative prior to over-smoothing, and to the final layer, which carries the most refined global context. The attention profile is neither uniform nor trivially concentrated on a single layer; instead, it forms a task-adapted weighting that balances local and global information. The embedding evolution plot (right) corroborates this: the normed difference between final-layer and intermediate representations grows with depth, confirming that early layers carry distinct information worth attending to. These visualizations demonstrate that HISTOGRAPH learns meaningful, non-trivial layer weightings that are consistent with the multi-scale information structure of deep GNNs.

## I    DISCUSSION OF PROTEINS PERFORMANCE

The large gain on PROTEINS (+16.6% over the next-best baseline) merits detailed discussion. The ablation study in Table 6 isolates the contribution of each HISTOGRAPH component: removing signed normalization drops accuracy to 74.45%, removing layer-wise attention to 78.61%, and removing node-wise attention to 80.78%—confirming that all three components are necessary. Table 16 further shows that uniform averaging (70.08%) and randomized attention (80.05%) fall far short, indicating the gain is driven by learned, signed layer weighting rather than any implementation artifact. We attribute the particularly strong performance on PROTEINS to the fact that protein graphs exhibit rich multi-scale structural information across GNN depths, which HISTOGRAPH's trajectory-aware readout is well-positioned to exploit. Protein classification benefits from combining local motifs (secondary structure elements captured by early GNN layers) with global fold topology (captured by deeper layers), and HISTOGRAPH's layer-wise attention naturally learns to weight both scales.

Table 15: Performance of HISTOGRAPH with different numbers of historical layers on PTC.

| #Historical Layers | PTC (%) |
|---|---|
| 5 | $79.1 \pm 4.8$ |
| 3 | $75.6 \pm 3.7$ |
| 1 | $73.8 \pm 4.3$ |

Table 16: Comparison between different aggregation options of all layers: mean over all layers, randomized attention, and HISTOGRAPH performance across datasets.

| Dataset | Mean over all layers | Randomized attention | HISTOGRAPH |
|---|---|---|---|
| IMDB-MULTI | $54.73 \pm 2.3$ | $54.73 \pm 4.3$ | $\mathbf{61.9 \pm 5.5}$ |
| IMDB-BINARY | $75.5 \pm 2.2$ | $76.6 \pm 1.4$ | $\mathbf{87.2 \pm 1.7}$ |
| PROTEINS | $70.08 \pm 5.8$ | $80.05 \pm 3.2$ | $\mathbf{97.8 \pm 0.4}$ |
| PTC | $73.24 \pm 3.2$ | $73.56 \pm 2.56$ | $\mathbf{79.1 \pm 4.8}$ |

## J INFERENCE TIME ANALYSIS

Table 18 reports average inference time (in milliseconds) for GCN backbones with 3 and 32 layers on MOLHIV and TOXCAST. Since inference involves no gradient computation, the FT and End-to-End modes yield identical runtimes. At 32 layers on MOLHIV, HISTOGRAPH is $\sim 4.8\times$ faster than GMT (3.174 ms vs. 15.158 ms) while adding only modest overhead over MeanPool (3.174 ms vs. 2.130 ms). Similar trends hold on TOXCAST, where HISTOGRAPH is $\sim 3.7\times$ faster than GMT at 32 layers. The overhead relative to MeanPool reflects the cost of the HISTOGRAPH head ($O(NLD + N^2 D)$), which remains small for the molecular-scale graphs evaluated here.

Table 17: Graph classification accuracy (%) ↑ across varying model depths, comparing methods over multiple datasets and approaches. The top three methods for each setting are marked by First, Second, and Third.

| Dataset | Method | Number of Layers | | | |
|---|---|---|---|---|---|
| | | 5 | 16 | 32 | 64 |
| IMDB-M | MeanPool | 54.0 | 54.7 | 52.0 | 52.0 |
| | FT | 67.3 | 64.7 | 57.3 | 66.0 |
| | Full FT | 58.0 | 62.7 | 58.0 | 57.3 |
| | End-to-End | 61.9 | 58.7 | 58.7 | 54.7 |
| IMDB-B | MeanPool | 76.0 | 76.0 | 76.0 | 71.0 |
| | FT | 94.0 | 84.0 | 81.0 | 78.0 |
| | Full FT | 94.0 | 81.0 | 81.0 | 81.0 |
| | End-to-End | 87.2 | 79.0 | 79.0 | 72.0 |
| PROTEINS | MeanPool | 75.0 | 75.0 | 75.0 | 75.9 |
| | FT | 97.3 | 77.7 | 75.9 | 77.8 |
| | Full FT | 97.3 | 84.8 | 80.4 | 97.3 |
| | End-to-End | 97.8 | 78.6 | 84.8 | 94.6 |
| PTC | MeanPool | 77.1 | 68.6 | 71.4 | 68.5 |
| | FT | 85.7 | 80.0 | 71.5 | 71.4 |
| | Full FT | 85.7 | 97.1 | 82.9 | 80.0 |
| | End-to-End | 79.1 | 88.6 | 85.7 | 80.0 |

Table 18: Average inference time (ms) per epoch for GCN backbones with 3 and 32 layers on MOLHIV and TOXCAST.

| Dataset | Layers | GMT | HISTOGRAPH (E2E / FT) | MeanPool |
|---|---|---|---|---|
| MOLHIV | 3 | 1.945 | 0.848 | 0.387 |
| | 32 | 15.158 | 3.174 | 2.130 |
| TOXCAST | 3 | 0.535 | 0.764 | 0.392 |
| | 32 | 10.678 | 2.859 | 1.773 |

