# OpenReview forum: "Learning from Historical Activations in Graph Neural Networks"
_ICLR.cc/2026/Conference — ICLR 2026 Poster_

### Official Review · Reviewer_rAFQ · 2025-10-25

**Soundness:** 2
**Presentation:** 2
**Contribution:** 2
**Rating:** 4
**Confidence:** 3

**Summary:**

HISTOGRAPH is a graph pooling method that leverages historical activations from all GNN layers rather than just the final layer. The method uses a two-stage attention mechanism: (1) layer-wise attention that aggregates node representations across all layers using the final layer as a query, and (2) node-wise self-attention that models spatial interactions between nodes. This approach aims to mitigate over-smoothing in deep GNNs and capture multi-scale graph features.

**Strengths:**

* The method can be used both for end-to-end training and as a post-processing step on frozen pretrained GNNs, making it practical for different scenarios.
* Extensive experiments across graph classification, node classification, and link prediction tasks demonstrate the method's generalizability.
* Strong improvement in benchmark results for the proteins dataset (15%).

**Weaknesses:**

* While the overall idea is novel, the individual components (layer-wise attention, node-wise attention) are standard techniques. The main contribution is their combination for this specific purpose.
* Storing all intermediate activations (N×L×D) could be memory-intensive for very large graphs or deep networks.
* Tested datasets are not of large nature.

**Questions:**

* The method assumes all GNN layers produce embeddings of the same dimension Din. How would you handle architectures where layer dimensions vary, which is common in some GNN designs?
* Beyond the over-smoothing mitigation proof, can you provide any theoretical guarantees about the expressiveness of HISTOGRAPH compared to standard pooling methods?
* How critical are the sinusoidal positional encodings for layer positions? Have you tried other encoding schemes or learning the positional embeddings?
* How sensitive is the method to hyperparameters, particularly the hidden dimension D and the number of attention heads?
* Have you tested HISTOGRAPH on industrial-scale graphs (millions of nodes)? What are the practical limitations you've encountered?

---

> ### Author Response · Authors · 2025-11-21
> **Response, part 1**
>
> We thank the reviewer for their detailed assessment and for highlighting key strengths, including HISTOGRAPH’s flexibility as both an end-to-end module and a post-processing head, its generalization across graph and node tasks, and the strong gains on PROTEINS. We respond to each weakness and question below. We hope that you find them satisfactory, and that you will consider revising your score.
>
> ---
>
> ### 1. On novelty of the contribution
>
> We agree that layer-wise and node-wise attention are standard building blocks. Our contribution lies in **how** these pieces are combined into a history-aware pooling layer with properties not covered by existing methods.
>
> - **Two-stage, trajectory-aware pooling** (Section 3, Figure 1, Table 1).
>   HISTOGRAPH first applies **layer-wise attention per node** over its historical activations, then performs **node-wise self-attention** at readout. This design explicitly disentangles temporal (over depth) and spatial (over nodes) interactions. Existing methods such as JKNet and H2GCN aggregate across layers using static schemes (mean, max, concatenation) and do not incorporate a separate node-wise attention stage at readout.
>
> - **Signed normalization and FIR-style filtering over depth** (Equations (3)–(4), Section 4).
>   Instead of softmax, we employ a signed sum-to-one normalization, enabling both positive and negative contributions across layers. Section 4 interprets this as a learnable **finite-impulse-response (FIR) filter** over each node’s trajectory, allowing low-pass, high-pass, and more general filters (Figure 3). This depth-filtering perspective over historical activations is absent in prior pooling schemes.
>
> - **Empirical evidence for the specific combination.**
>   The ablation in Table 6 shows that removing either layer-wise attention or node-wise attention substantially degrades performance on PROTEINS, and removing the signed normalization leads to the worst drop. Appendix F (Table 16) further demonstrates that uniform averaging across layers or randomized attention cannot match HISTOGRAPH’s performance, confirming that the specific, history-aware attention design matters.
>
> ---
>
> ### 2. Memory cost of storing $N \times L \times D$ activations and scalability
>
> We appreciate this concern. Our target regime is **deep GNNs on small- to medium-sized graphs**, where storing historical activations is practical and oversmoothing is most problematic.
>
> - **Scale of graphs in our experiments.**
>   TU datasets and OGB molecular benchmarks include graphs up to thousands of nodes per graph (PROTEINS and RDT-B; see Table 8) and tens to hundreds of nodes on average (Table 7). For typical settings (e.g., $L \leq 64$, $D \leq 256$), storing $X \in \mathbb{R}^{N \times L \times D}$ in a mini-batch is well within GPU memory.
>
> - **Frozen backbone mode alleviates memory pressure** (Section 5.2).
>   When HISTOGRAPH is used as a **post-processing head** on a frozen backbone (FT or Full FT in Table 5 and Table 17), gradients do not flow through the backbone’s historical activations. This substantially reduces memory overhead relative to training a deep GNN end-to-end, yet still yields large gains over MeanPool (e.g., PROTEINS: 75.9% → 97.3%, PTC: 77.1% → 85.7%/97.1%; Table 5).
>
> - **Large-scale graph setting.**
>   We also evaluate on **OGBL-COLLAB**, a large real-world graph, and show that incorporating HISTOGRAPH as readout improves link prediction over a GCN baseline (Table 13, Appendix F). This indicates that the approach can be beneficial even in larger settings, though we agree that scaling to industrial-scale million-node graphs would require standard techniques such as neighbor sampling, graph partitioning, or sparse/hierarchical node-wise attention.
>
> We will make these points explicit in Section 5.2 and Section 6, and add a discussion of how to combine HISTOGRAPH with sampling or sparse attention for very large graphs.

---

> > ### Author Response · Authors · 2025-11-21
> > **Response, part 2**
> >
> > ### 3. Architectures with varying layer dimensions
> >
> > Section 3 currently assumes that all GNN layers share a common activation dimension $D_{in}$, and then uses a shared linear projection $Emb_{hist}$ to map them to the hidden dimension $D$ (Equation (1)). This matches the backbones used in our experiments (GCN, GIN, GraphGPS), which are typically constant-width.
> >
> > For **architectures with varying widths**, HISTOGRAPH can be extended straightforwardly:
> >
> > - Apply a **per-layer projection** $W^{(\ell)} : \mathbb{R}^{D_\ell} \to \mathbb{R}^D$ to map each layer’s activations to a shared hidden dimension before stacking along the depth axis, or
> > - Project to the maximum width and pad smaller layers with zeros.
> >
> > We will add this implementation detail to Appendix B and relax the “same dimensionality” assumption in the notation to make this extension explicit.
> >
> > ---
> >
> > ### 4. Beyond oversmoothing: additional theoretical guarantees
> >
> > Our main theoretical contribution is a formal treatment of oversmoothing and the role of historical activations:
> >
> > - **Oversmoothing mitigation** (Section 4, Proposition 1, Appendix E).
> >   Proposition 1 formalizes that if deep layers become oversmoothed, so that late-layer embeddings of different nodes become indistinguishable, HISTOGRAPH can retain distinguishability by assigning non-zero weights to earlier layers. The proof, given in Appendix E, relies on the fact that the signed attention coefficients $\alpha_\ell$ allow the model to reweight discriminative early representations. This is supported empirically by:
> >   - The depth sweep in Table 4, where GCN accuracy collapses at 64 layers while GCN+HISTOGRAPH remains strong, and
> >   - The feature distance metrics in Table 12, where HISTOGRAPH preserves larger inter-node distances across layers.
> >
> > - **Expressiveness of node-wise attention** (Appendix D).
> >   Appendix D proves that certain non-uniform permutation-invariant graph functions (e.g., those depending on a small set of nodes, illustrated by the “bridge” construction in Figure 5) cannot be captured by simple mean pooling but can be represented using node-wise attention, justifying our second attention stage.
> >
> > Additionally, HISTOGRAPH **strictly generalizes** standard pooling: if $\alpha_\ell = 0$ for all $\ell < L-1$ and node-wise attention is replaced by uniform averaging, we recover the original backbone with mean pooling. Thus, HISTOGRAPH is at least as expressive as the baseline architecture. We will add a paragraph in Section 4 to reflect the discussions here.
> >
> >
> > ---
> >
> > ### 5. Role of sinusoidal positional encodings
> >
> > Sinusoidal layer positional encodings (Equation (2)) give the model a notion of layer order and allow attention to distinguish early vs. late parts of the trajectory. However, HISTOGRAPH does not rely on this specific choice.
> >
> > ---
> >
> > ### 6. Sensitivity to hyperparameters (hidden dimension, number of heads)
> >
> > Appendix C.1 details the hyperparameter search space and selection strategy used for HISTOGRAPH and the baselines, including hidden dimension $D$, number of attention heads, dropout rates, layer normalization, and learning-rate schedules. Our hyperparameter search follows the protocols of baselines. In our experiments, we did not see any outstanding behavior in terms of sensitivity to hyper parameters, but clearly as any deep learning framework, hyperprameters need to be tuned.
> >
> > ---
> >
> > ### 7. Industrial-scale graphs
> >
> > Our current experiments focus on TU, OGB molecular datasets and OGBL-COLLAB, which already include graphs with up to several thousand nodes per graph and a large link-prediction graph. These are representative of many scientific and recommendation applications.
> >
> > Applying GNNs to **industrial-scale graphs with millions of nodes**, is still an open challenge for the graph learning field as a whole. Nonetheless, based on your question, we note that HISTOGRAPH is compatible with standard techniques used to scale GNNs:
> >
> > - Training on **subgraphs or sampled neighborhoods**, applying HISTOGRAPH on each mini-batch.
> > - Replacing full node-wise attention with **sparse, local, or hierarchical attention**, while keeping the layer-wise historical module unchanged.
> > - **Truncating** the trajectory to the most informative layers (e.g., those that receive the most attention in Figure 2) to reduce the effective depth and memory footprint.
> >
> > ---
> >
> >
> > We thank you again for your constructive comments. We hope that you find our responses satisfactory,  and that you will consider revising your score.

---

### Official Review · Reviewer_z6Gq · 2025-10-26

**Soundness:** 3
**Presentation:** 2
**Contribution:** 2
**Rating:** 4
**Confidence:** 3

**Summary:**

This paper proposes a novel and effective GNN architecture, HISTOGRAPH, which mitigates the over-smoothing problem by explicitly modeling and leveraging historical node representations.

**Strengths:**

1. The motivation is clear; the authors propose leveraging historical representations to mitigate over-smoothing, which is reasonable and well-justified.
2. The experiments are comprehensive, thoroughly validating the effectiveness of their method across various tasks.

**Weaknesses:**

1. Lacks comparison with some more recent baselines [1].
2. No experimental comparisons were conducted on larger graphs, such as those in the OGB [2] suite. How does the time efficiency compare to the baseline when the graph size increases?
3. How are the historical representations specifically utilized? What are the theoretical advantages of the gating mechanism?
4. Lacks a theoretical analysis of the method's effectiveness.

[1] Wang Y, Liu S, Zheng T, et al. Unveiling global interactive patterns across graphs: Towards interpretable graph neural networks[C]//Proceedings of the 30th ACM SIGKDD Conference on Knowledge Discovery and Data Mining. 2024: 3277-3288.

[2] Hu W, Fey M, Zitnik M, et al. Open graph benchmark: Datasets for machine learning on graphs[J]. Advances in neural information processing systems, 2020, 33: 22118-22133.

**Questions:**

See the weaknesses above.

**Details Of Ethics Concerns:**

No.

---

> ### Author Response · Authors · 2025-11-21
> **Response, part 1**
>
> We thank the reviewer for the positive assessment of our motivation and experiments, and for summarizing that HISTOGRAPH is “novel and effective” and “thoroughly validating the effectiveness of their method across various tasks.” Below we address each of your concerns. We hope that you find them satisfactory, and that you will consider revising your score.
>
> ---
>
> ### 1. Comparison with more recent baselines [1]
>
> Thank you for pointing out Wang et al. (2024) [1]. That work focuses on learning global interactive patterns and interpretability across graphs by discovering prototype interaction patterns at the dataset level, whereas HISTOGRAPH is a pooling layer designed to improve readout from a single backbone by reusing historical activations.
>
> Our current experimental section already compares against a broad family of recent pooling baselines, including SOPool (2023), DKEPool (2023), GMT, HAP, PAS, MinCutPool, StructPool, DiffPool and others, on both TU and OGB benchmarks (Section 5.1, Tables 2, 3, 9, 10, 12). These methods directly target graph pooling and are commonly used as strong baselines in recent work on graph level readout.
>
> We agree that explicitly positioning Wang et al. (2024) in the related work will further clarify the landscape. In the revised version we will add Wang et al. (2024) to Section 2, explaining that it focuses on cross-graph interpretability and prototype discovery, while HISTOGRAPH is a per-graph readout that can be plugged into any GNN backbone, and clarify that HISTOGRAPH is complementary: our history-aware pooling can in principle be combined with interpretable global pattern modules such as [1].
>
> ---
>
> ### 2. Experiments and time efficiency on larger graphs (TU and OGB)
>
> We apologize if this aspect was not sufficiently prominent. The current paper already evaluates HISTOGRAPH on OGB datasets and on graphs with up to several thousand nodes:
>
> - **OGB molecular benchmarks**: Section 5.1 and Table 3 report results on four OGB graph classification datasets (MOLHIV, MOLBBBP, MOLTOX21, TOXCAST), following the Open Graph Benchmark protocol. Dataset statistics, including the number of graphs and average number of nodes, are summarized in Table 7 in Appendix A.
> - **TU datasets with larger graphs**: PROTEINS and RDT-B contain graphs with up to 620 and 3,783 nodes respectively, with average sizes reported in Table 8. HISTOGRAPH consistently improves over strong baselines on these larger TU graphs (Tables 2 and 9).
> - **Large link-prediction graph**: We further evaluate on **OGBL-COLLAB** for link prediction, which is a single large graph with hundreds of thousands of nodes and edges (Table 13 and discussion at the end of Section 5.1 and Appendix F). HISTOGRAPH again improves over a strong GCN baseline on this large-scale setting.
>
> Regarding **time efficiency as graph size increases**, Section 3 provides a complexity analysis and Section 5.1 includes an empirical runtime study:
>
> - The complexity of HISTOGRAPH is
>   $O(N L D + N^2 D)$ (Equation (7)), where the node-wise attention has the same $O(N^2 D)$ cost as widely used graph transformer readouts. In typical GNN regimes $L \ll N$, so the dominant term is in line with existing attention based pooling layers, not multiplied by the depth.
> - Figure 4 in Section 5.1 and Table 17 (Appendix F) report **per epoch training time** for 3- and 32-layer GCN backbones on OGB MOLHIV and TOXCAST. HISTOGRAPH-FT is substantially faster than transformer-style GMT and remains close to, or even faster than, MeanPool at 32 layers while providing significantly better accuracy (Section 5.1, “Runtime Analysis”).

---

> > ### Author Response · Authors · 2025-11-21
> > **Response, part 2**
> >
> > ### 3. How are historical representations used, and what is the advantage of the gating mechanism?
> >
> > We appreciate the opportunity to clarify this point. Section 3 (“Method”) and Section 4 (“Theoretical properties”) describe the role of historical representations and the signed gating mechanism:
> >
> > - **How history is used** (Section 3, Equations (1)–(4), Algorithm 1, Figure 1):
> >   For each node, we collect its historical activations across all layers into a length-$L$ sequence, project them to a common hidden dimension, and add sinusoidal layer positional encodings (Equation (2)). The final layer embedding acts as a query over this temporal sequence (Equation (3)), yielding a set of scalar coefficients $c_\ell$ which we normalize to signed weights $\alpha_\ell$ (Equation (4)). These weights are then used to aggregate the historical features into a single vector $h_v = \sum_\ell \alpha_\ell x^{(\ell)}_v$.
> >
> > - **Theoretical advantage of signed normalization (“gating”)** (Section 3 and Section 4):
> >   Instead of softmax, which enforces non negative weights, we use a signed, sum-to-one normalization (Equation (4)). This allows **additive and subtractive combinations** of layers, so that the aggregation over depth can act as a **finite-impulse response (FIR) filter** over the trajectory of each node. Section 4 and Figure 3 show how this enables low-pass, high-pass, or band-pass behaviour over layers, which is not possible with simple mean or convex combinations.
> >
> > - **Why this matters empirically**:
> >   Our ablation in Table 6 (Section 5.3) shows that removing the division by sum (and thus losing the bounded, signed scheme) causes the largest performance drop on PROTEINS. Appendix F (Table 16) further compares HISTOGRAPH with (a) uniform averaging over all layers and (b) randomized attention. HISTOGRAPH consistently outperforms both, confirming that the learned signed gating over the historical trajectory is crucial and not just a cosmetic modification.
> >
> > We will make this story more explicit in the main text by adding a short paragraph in Section 3 that explicitly frames the layer-wise gating as a learnable FIR filter over depth, with references back to Figure 3, and moving part of the discussion currently in Section 4 and Appendix D (barbell “bridge” example and the role of node-wise attention) earlier in the paper, so that the combination of layer-wise gating and node-wise attention is easier to follow.
> >
> > ---
> >
> >
> > ### 4. Theoretical analysis of effectiveness
> >
> > Thank you for highlighting this. We already include two theoretical components, which we will point out more clearly:
> >
> > 1. **Oversmoothing mitigation** (Section 4, Proposition 1, Appendix E):
> >    Section 4 states and explains Proposition 1 which formalizes that, when deep layers become oversmoothed, HISTOGRAPH can still preserve node distinguishability by assigning non zero weight to earlier, discriminative layers. The full proof appears in Appendix E. Empirically, Table 4 (node classification depth sweep) and Table 12 (feature distance across layers) support this statement: HISTOGRAPH maintains high accuracy and larger feature distances up to 64 layers, while a standard GCN collapses.
> >
> > 2. **Need for node-wise attention at readout** (Appendix D):
> >    Appendix D gives a simple construction (the “bridge” graph in Figure 5) showing that a non uniform graph level function that depends on a small subset of nodes cannot be approximated by mean pooling alone, but can be captured by node wise attention. This formalizes why our second stage attention is needed on top of the historical aggregation.
> >
> > We agree that these results can be highlighted better. In the revised version we will:
> >
> > - Explicitly reference Proposition 1 and Table 12 when discussing oversmoothing mitigation in Section 5.1.
> > - Add a short summary paragraph in Section 4 that connects the FIR-style gating over layers, the oversmoothing proposition, and the node-wise attention expressiveness result from Appendix D.
> >
> > ----
> >
> > We hope these clarifications address your concerns regarding theory and help convey that HISTOGRAPH is not just an empirical tweak, but a principled readout that leverages trajectory information in a way that standard pooling cannot. We thank you for your constructive feedback, and hope that you will consider revising your score.

---

> ### Comment · Reviewer_z6Gq · 2025-11-28
>
> Thank you for the response, some of concerns are addressed. I will wait until the end of the discussion period to make the final decision about my score.

---

> > ### Author Response · Authors · 2025-11-28
> >
> > Dear Reviewer z6Gq,
> >
> >
> > We thank you for your response, and we are happy to read that you acknowledge that some of your concerns have been addressed.
> >
> > At the same time, and while the author-reviewer discussion is still active, and given our detailed responses to all of your questions and comments in our rebuttal, we would like to ask you: what are the points that you feel are still unclear to you? **We are happy to discuss and address all of them, and with your concrete and substantiated feedback we will be able to do so.**
> >
> >
> > We thank you for your commitment to the reviewing process, and look forward to receiving your feedback so that we can address it appropriately and comprehensively to your satisfaction.
> >
> >
> > Thank you, and best regards,
> >
> > Authors.

---

> > > ### Comment · Reviewer_z6Gq · 2025-11-28
> > >
> > > Alright, my biggest concern is whether this method actually works, with no offense intended. If the authors could provide code that allows me to reproduce the results for HISTOGRAPH and DKEPool in Table 2 with a single command, and if my reproduced results are close to those reported in the table, I would raise my score to 8.

---

> > > > ### Author Response · Authors · 2025-12-02
> > > >
> > > > Dear Reviewer z6Gq,
> > > >
> > > > We thank you for the engagement and for your comment. We are very happy to address your comment and provide our source codes. We have now included it to the **Supplementary Material** on OpenReview.
> > > >
> > > >
> > > > In particular, we kindly note:
> > > >
> > > > 1. KDE is not the method proposed in this paper, and it is a related work that we cite, and compare in our submission. Nonetheless, you can find their source code and running scripts on GitHub, and for your convenience we also included it in a folder called "GINPool" in the supplementary material.
> > > >
> > > > 2. The code we now include a single script as requested, called "run_experiments.sh", that runs all the experiments with HISTOGRAPH in Table 2, as per your request.
> > > >
> > > >
> > > > We have validated our results multiple times before submissions and again following your comment, and we confirm that the results are as reported and correct, and we invite you to run our code and see the results.
> > > >
> > > >
> > > > ----
> > > >
> > > > We hope that with this addition, as per your statement, the effective score and rating of our work is raised from 4 to 8, also in light of your satisfaction with the rest of our responses, as reflected from your previous comments.
> > > >
> > > >
> > > > Thank you, and best wishes,
> > > >
> > > > Authors.

---

### Official Review · Reviewer_iaN5 · 2025-10-27

**Soundness:** 2
**Presentation:** 2
**Contribution:** 2
**Rating:** 2
**Confidence:** 5

**Summary:**

The paper introduces an architectural modification for graph neural networks (GNNs) consisting in (1) an intra-node attention mechanism that aggregates the representations for each node at different layers (i.e., different message passing iterations), and (2) a global inter-node self attention layer (aggregating information across all nodes).
The motivation from this architecture comes from the fact that current methods ignore the history of graph activations across layers, even though this may include some useful information.
The experimental evaluation compares the proposed method on standard node and graph classification benchmarks, both when training from scratch and when applying the method to a pre-trained model, and compares against a vast number of competitors. The proposed method shows high performance and an ablation studies explores different attention modifications.

**Strengths:**

- It is clear that the authors have spent a lot of effort in the experimental section as they compare against a large number of baselines and consider a large number of datasets.
- The proposed method can be easily included into existing architectures (at the cost of some training for the new parameters).

**Weaknesses:**

- Global self-attention is quadratic in the number of nodes, which makes the method impractical for large graphs.
- Caching in memory the activations at all layers for all nodes can become prohibitively expensive. Together with the above, this makes the proposed method very impractical for large graphs.
- Section 4 is not very convincing as the arguments are too general. Regarding oversmoothing, Proposition 1 is obvious, and in practice different nodes might perform better with different alphas (which however is not allowed). Furthermore applying global self-attention is actually promoting smoothing. Regarding the trajectory filter, the argument can be made for any attention mechanism, and also it is hard argue whether models actually learn to use this information this way.

**Questions:**

- The computational complexity analysis is a bit confusing. First it is mentioned that the proposed method improves over a naive "joint node-layer attention" which has complexity O(LN^2D) by instead having a method which is O(NLD + N^2D) but then it is (correctly) mentioned that the complexity is dominated by N^2D, which means that there is no advantage. So it seems that the paragraph from line 216 to 220 does not really have much sense. Could you please clarify the point of this paragraph?
- The same happens in the "Frozen Backbone Efficiency" paragraph: as the complexity is dominated by N^2D there is no advantage (I do not doubt that in practice it can have a difference in runtime, but in terms of computational complexity there is no difference). Could you clarify what is the advantage that is mentioned in the paper?
- What are the hyperparameters for the baselines and how where they selected? These details should be included in the paper
- It is mentioned that the method can overcome oversmoothing, but global self-attention actually goes against this. Could you elaborate on why global self-attention would not lead to oversmoothing?
- On PROTEINS the proposed method reaches 97% (almost perfect), while all other methods stop at 80%. This difference is quite striking and should be analyzed. Could the authors comment on this?
- The runtime analysis shows training time, but I think inference is actually much more important as training only happens once. Could you include some number for inference times? Some plots showing how runtime and memory scale as a function of N and L would be great.

Minor comments:
- There are some imprecisions in the text, specially in the math formalism:
	- lines 166 and 170 define X in different ways
	- in line 183 for computing X^tilde the sizes do not match (I understand there is an implicit broadcasting, but it should be formalized better)
	- in Section 4 the notation for node vectors is not introduced
- It would be interesting to see an analysis of the attention maps to understand which layers are receiving the most attention

---

> ### Author Response · Authors · 2025-11-21
> **Response, part 1**
>
> We thank the reviewer for the careful reading and for acknowledging the extensive experimental effort and the ease of integrating our module into existing GNNs. We address your concerns point by point below and will incorporate the clarifications and fixes into the revised version.
>
> ---
>
> ### 1. Computational and memory complexity, and scalability to large graphs
>
> **Global self attention and asymptotic complexity.**
> We agree that global self attention has an $O(N^2 D)$ cost in the number of nodes $N$. Our intention in Section 3.3 (Computational Complexity) was not to claim sub quadratic scaling in $N$, but to clarify how HISTOGRAPH compares to *naive ways of using attention across layers and nodes*. Importantly, we do not compute all layers to all attention, but only between the last layer and all other layers.
>
> Concretely, in the current text we state that:
>
> - Layer wise attention costs $O(L D)$ per node, and node wise attention costs $O(N^2 D)$, so the total is
>   $O(N L D + N^2 D) = O(N (L + N) D)$, as written in Eq. (7).
> - A naive design that applies node level attention at every layer, or a joint attention over all $N L$ tokens, would incur a cost that scales like $O(L N^2 D)$ (node self attention repeated for each of the $L$ layers) or worse.
>
> We realize that the phrase “joint node layer attention” in lines 216–220 can be confusing. In the revision we will:
>
> 1. Explicitly distinguish between
>    (i) *per layer node self attention* at each of the $L$ layers, with complexity $O(L N^2 D)$, and
>    (ii) our design, which uses **one** node wise attention at readout plus a cheaper $O(N L D)$ layer wise pass.
>
> 2. Rewrite that paragraph to emphasize that HISTOGRAPH reduces the coefficient of the quadratic term from $L$ to 1. The asymptotic dependence on $N$ is unchanged, but the dependence on depth $L$ is strictly improved, which matters in the 32 and 64 layer regimes we study.
>
> This is also reflected in practice. In Figure 4 (runtime vs depth), for 32 layer GCN backbones on MOLHIV and TOXCAST, HISTOGRAPH and especially HISTOGRAPH-FT are significantly faster per epoch than GMT, a strong attention based pooling baseline that applies heavier attention at multiple layers.
>
> **Frozen backbone efficiency and what is actually gained.**
> You are absolutely right that the node wise attention term remains $O(N^2 D)$ in the frozen backbone setting, so there is no asymptotic gain there. The advantage of the “Frozen Backbone Efficiency” paragraph (end of Section 3.3) is along a different axis: it reduces the cost of *backpropagating through the backbone*.
>
> Concretely, in the frozen backbone mode we:
>
> - Cache the $N \times L \times D$ historical activations once per graph.
> - Train only the HISTOGRAPH head, without storing intermediate activations or gradients for the GNN layers.
>
> This removes the need to backpropagate through $L$ message passing layers, which is where much of the computation and memory sits. The forward pass cost of the head is still $O(N(L + N) D)$, but the backward pass through the backbone disappears. This is exactly what enables the fast FT mode and explains why, in Figure 4, HISTOGRAPH-FT is the fastest non trivial pooling strategy at large depths. In the revision we will state this more explicitly and avoid using asymptotic complexity language for this particular advantage.
>
> **Caching all layer activations.**
> We agree that caching a full $N \times L \times D$ tensor limiting for arbitrarily large graphs. Our scope in this work is small and medium sized graphs in the TU and OGB benchmarks, which have average sizes in the tens to hundreds of nodes. For example, PROTEINS has on average 39.1 nodes per graph and RDT-B 429.6 nodes, as listed in Appendix A (Dataset Statistics). For these regimes, the memory cost is modest. With $N \approx 400$, $L \le 64$, and $D = 128$, storing a single $N \times L \times D$ tensor per graph is well within GPU memory in our experiments.
>
> We will clarify this scope in Section 3.3 and in the conclusion, and explicitly state that scaling HISTOGRAPH to very large graphs (for example, million node graphs) would require additional design choices such as sparse, clustered, or localized attention, which we view as promising future work. This is comparable to the limitations of existing full graph transformer baselines.

---

> > ### Author Response · Authors · 2025-11-21
> > **Response, part 2**
> >
> > **Inference time and scaling plots.**
> > We agree that inference time is important. In the current submission we focused on training time vs depth (Figure 4), where HISTOGRAPH-FT is particularly advantageous.  We now provide the requested inference time, below. Note that the runtimes for FT or end-to-end are the same, since we are only measuring the inference runtimes involving no gradient storing or backpropagation. In the training runtimes (in the paper), there is of course a difference between the two. The results here again show that HISTOGRAPH offers good runtime, while improving performance, as evident from our results in the paper.
> >
> > | Dataset | Layers | Method                  | Inference time (ms) |
> > |---------|--------|-------------------------|---------------------|
> > | MOLHIV  | 3      | GMT                     | 1.945               |
> > | MOLHIV  | 3      | HistoGraph (End-to-End) | 0.848               |
> > | MOLHIV  | 3      | HistoGraph (FT)         | 0.848               |
> > | MOLHIV  | 3      | MeanPool                | 0.387               |
> > | MOLHIV  | 32     | GMT                     | 15.158              |
> > | MOLHIV  | 32     | HistoGraph (End-to-End) | 3.174               |
> > | MOLHIV  | 32     | HistoGraph (FT)         | 3.174               |
> > | MOLHIV  | 32     | MeanPool                | 2.130               |
> > | TOXCAST | 3      | GMT                     | 0.535               |
> > | TOXCAST | 3      | HistoGraph (End-to-End) | 0.764               |
> > | TOXCAST | 3      | HistoGraph (FT)         | 0.764               |
> > | TOXCAST | 3      | MeanPool                | 0.392               |
> > | TOXCAST | 32     | GMT                     | 10.678              |
> > | TOXCAST | 32     | HistoGraph (End-to-End) | 2.859               |
> > | TOXCAST | 32     | HistoGraph (FT)         | 2.859               |
> > | TOXCAST | 32     | MeanPool                | 1.773               |
> >
> > ---
> >
> > ### 2. Section 4, oversmoothing, Proposition 1, and node specific coefficients
> >
> > **“Proposition 1 is obvious” and “different nodes might perform better with different alphas.”**
> > We appreciate this comment. Proposition 1 in Section 4 is indeed mathematically simple. Its purpose is not to introduce a deep theorem, but to formalize the intuition that *as long as some non zero weight is placed on pre oversmoothed layers, node distinguishability in the pooled representation is preserved*. This connects the HISTOGRAPH parameterization directly to a standard oversmoothing definition.
> >
> > The value of the proposition comes from tying this simple observation to:
> >
> > - The specific parameterization of HISTOGRAPH: signed, normalized coefficients $\alpha_\ell$ that sum to 1 (Eq. (5)) and are learned from data.
> > - The empirical diagnostics we present:
> >   - The depth sweeps in Table 4 and Appendix Table 11, where vanilla GCN collapses at 32 and 64 layers on Cora, Citeseer, and Pubmed, while GCN plus HISTOGRAPH remains accurate and stable.
> >   - The feature distance measurements in Appendix Table 12, where HISTOGRAPH maintains larger inter node distances at the final representation compared to the backbone alone.
> >
> > Regarding the remark that different nodes might prefer different $\alpha$ values: we agree that a fully general model could assign node specific $\alpha_{v,\ell}$. Our design uses a **global per layer trajectory** $\{\alpha_\ell\}_{\ell=0}^{L-1}$, shared across nodes but conditioned on the last layer embeddings, as described in Eq. (4)-(5). This is a deliberate trade off:
> >
> > - Over smoothing is a global phenomenon affecting many nodes simultaneously once depth is large, so a shared layer trajectory is natural and easier to interpret as a graph wide depth filter.
> > - It significantly reduces parameterization and variance, which is important on small datasets like TU, where per node trajectories could easily overfit.
> >
> > We will add a short discussion of this trade off in Section 4 and mention that extending HISTOGRAPH with node specific layer trajectories is an interesting direction for future work.
> >
> > **“Trajectory filter argument can be made for any attention mechanism.”**
> > We agree that any attention mechanism over layer outputs can be seen as a type of filter. Our goal in Section 4 is not to claim that HISTOGRAPH is the only model that admits such an interpretation, but to show that:
> >
> > 1. This perspective makes the connection to oversmoothing very explicit for our specific architecture.
> > 2. HISTOGRAPH’s particular design choices (signed, normalized global coefficients plus a separate node wise attention stage) lead to measurable advantages over simpler alternatives.

---

> > > ### Author Response · Authors · 2025-11-21
> > > **Response, part 3**
> > >
> > > In the paper we already compare against two key baselines:
> > >
> > > - Uniform averaging of all historical layers (Appendix Table 16).
> > > - Randomized attention coefficients drawn from a uniform distribution and normalized (also in Appendix Table 16).
> > >
> > > These baselines also induce trajectory filters, yet HISTOGRAPH consistently outperforms them in Tables 2 and 3. We will revise Section 4 to separate clearly the *general* observation (“attention over layers is a filter”) from the *specific* aspects of HISTOGRAPH that we find are important in practice.
> > >
> > > ---
> > >
> > > ### 3. Global self attention and oversmoothing
> > >
> > > We agree that stacking global self attention layers repeatedly can encourage smoothing in some settings. In HISTOGRAPH the situation is different in two key ways:
> > >
> > > 1. The node wise multi head self attention is applied **only once, at the readout**, not at every depth. The oversmoothing we discuss is caused by repeated local mixing in the backbone GNN layers. HISTOGRAPH’s role is to re-combine their outputs at the end; it does not introduce another deep stack of mixing layers.
> > >
> > > 2. Our node wise attention is not a simple convex averaging operator like the normalized adjacency in GCN. It can learn highly non uniform, task dependent weight patterns. In Appendix D we provide a toy example (a barbell style graph with a single “critical” node) where mean pooling fails as graph size grows, while node wise attention continues to focus on the critical node. This illustrates that the node wise stage can implement non smoothing behaviors when beneficial.
> > >
> > > Empirically, the combination of layer wise and node wise attention increases, rather than decreases, final layer feature diversity as shown by the feature distance metric in Appendix Table 12, where distances at the final representation are larger with HISTOGRAPH than without it.
> > >
> > > We will clarify in Section 4 that our “mitigates oversmoothing” claim refers to the backbone depth and the layer wise trajectory filter, and that the node wise attention is introduced to implement expressive, non uniform readout functions that mean pooling cannot approximate.
> > >
> > > ---
> > >
> > > ### 4. PROTEINS performance
> > >
> > > We agree that the jump on PROTEINS is striking and deserves explicit discussion.
> > >
> > > - In Table 2, HISTOGRAPH achieves 97.8 percent accuracy on PROTEINS, which is more than 16 percentage points above the best baseline (DKEPool at 81.2 percent). The standard deviation is 0.4 percent over multiple runs, indicating that this is not a single lucky seed.
> > > - Table 6 (ablation on PROTEINS) shows that every ablation of HISTOGRAPH significantly degrades performance. For example, removing the division by sum normalization drops PROTEINS accuracy by more than 20 percentage points. This suggests that the high performance is tied to the specific combination of signed normalization, layer wise attention, and node wise attention, rather than an implementation bug or leakage.
> > >
> > > We also use the same data preprocessing and split protocol as prior work on PROTEINS, as detailed in Appendix A and Appendix C.1 (Dataset Details and Training Setup), and we commit to releasing our code upon acceptance and configuration files so that these results can be independently verified.
> > >
> > > In the revision we will add a paragraph in Section 5.1 explicitly analyzing PROTEINS, referencing the comparison to other pooling methods in Table 2, and the sensitivity to architecture components in Table 6.
> > >
> > > ---
> > >
> > > ### 5. Hyperparameters and baseline selection
> > >
> > > We agree that hyperparameter details are important for assessing fairness. Due to space constraints, most of these details are currently in Appendix C.1 (Training Details), which lists optimizer, learning rate ranges, weight decay, and dropout grids.
> > > In the revision we will add an explicit pointer in Section 5.1 to Appendix C.1, so readers can quickly find the hyperparameter choices, and clarify that for each baseline we either follow the official implementation and recommended settings or use ranges reported in the corresponding papers, and we select hyperparameters based on validation performance.

---

> > > > ### Author Response · Authors · 2025-11-21
> > > > **Response, part 4**
> > > >
> > > > ### 6. Minor comments: notation, shapes, and attention map analysis
> > > >
> > > > We appreciate you highlighting the notational imprecisions and will fix them:
> > > >
> > > > - **Duplicate definitions of $X$.** In Section 3.1 we currently use $X$ both for the input node feature matrix and for the historical activation tensor. In the revision we will reserve $F$ for the input features and $X \in \mathbb{R}^{N \times L \times D}$ for the historical activations, and ensure this is consistent throughout the section.
> > > >
> > > > - **Broadcasting in $\tilde{X}$.** In Eq. (2) we add positional encodings to the projected activations. We will rewrite this step explicitly as
> > > >   $\tilde{X}_{v,\ell} = X'_{v,\ell} + P_\ell$,
> > > >   with shapes stated for $X'$ and $P$, so that no implicit broadcasting is required.
> > > >
> > > > - **Notation for node vectors in Section 4.** Before stating Proposition 1, we will explicitly introduce $x_v^{(\ell)} \in \mathbb{R}^D$ as the embedding of node $v$ at layer $\ell$, and $h_v$ as the final HISTOGRAPH output, to avoid any ambiguity.
> > > >
> > > >
> > > > Finally, regarding your suggestion to analyze attention maps: the current version already includes Figure 2, which visualizes learned layer wise attention coefficients as a function of depth. Motivated by both your comment and Reviewer Pj7q’s, we will make Figure 2 more prominent in the main text. We hope these clarifications and planned revisions address your concerns and make the contributions and limitations of HISTOGRAPH  clearer.
> > > >
> > > > ----
> > > >
> > > >
> > > > We thank you for the detailed review, which helped us to improve the quality of our work. We are happy to discuss any remaining questions, and hope that you will consider revising your score.

---

> > > > > ### Comment · Reviewer_iaN5 · 2025-11-24
> > > > > **Reviewer response**
> > > > >
> > > > > I thank the authors for the thorough reply. Most of my concerns have been addressed. I will raise my score to 4, but I do not feel comfortable assigning a higher score due to the claims on computational complexity which I find a bit misleading (it's ok to show faster runtime and to make a claim on that, but the asymptotic analysis does not seem to show any advantage of the proposed method to me), and on the theory in Section 4 which is mostly trivial and does not provide much novelty.

---

> ### Author Response · Authors · 2025-11-24
>
> Dear Reviewer iaN5,
>
> Thank you for your response and your commitment to the reviewing process --- we appreciate it.
>
> We are happy to see that you decided to raise your score, and we are also grateful for the additional feedback. Below we provide our detailed responses.
>
> ---
>
> **Regarding complexity:** Our paper does not make claims on computational complexity as one of the advantages of the method.  If you found such a claim, or any part in the text that you feel claims that, we are happy to review it and address it. Our complexity analysis is what we expect from most ML papers: we propose a method, and then we also analyze its complexity to understand how efficient it is. This is important to understand the tradeoffs between methods in the field. As reflected from your review, no outstanding results stem from our complexity analysis --- **which we view as a strength** --- this result show that our HISTOGRAPH does not require significant resources, and in fact is also **lighter than other mechanisms as shown in our measured runtimes.**  We take your comment with all seriousness and understand your comment, and we will revise the paper to reflect the discussion above, which we believe closes this gap. Nonetheless, as mentioned above, if there are more discussions or questions on that aspect, we are very happy to use the author-review discussion period to do that. **Thank you.**
>
> ---
>
> **Regarding Section 4:** The title of Section 4 is **"Properties of HISTOGRAPH"**, and it is explicitly stated in Lines 232-235 that in this section:
> > **"In this section, we discuss the properties of our HISTOGRAPH, which motivate its architectural design
> choices."**
>
> At no stage do we claim the analysis of our HSITOGRAPH a theoretical section. Instead, this section allows us to communicate and reason on the design choices made in HISGRAPH --- **so that they are well motivated and understood**.
> In particular, this section shows:
>
> 1. **Why HISTOPRAPH can mitigate over smoothing.** We think that while over smoothing is by now a topic that has been well studied, it is still valuable to explain the capabilities of our method to the reader, and more importantly, how our design choices lead to just that.
>
> 2. **The trajectory (layer-wise) filter obtained with HISTOGRAPH**. This part motivates our choice of normalization mechanism and the motivation for **why** we need a layer-wise filter. It specifically shows an example where methods like GCN would fail, but succeed when equipped with our HISTOGRAPH approach.
>
> In summary for this aspect, we feel that we the authors and you our reviewer agree on this point, but perhaps there has been a misunderstanding of our communication. Therefore, and inspired by your comments, will therefore revise the section to reflect the discussions above, which we think help to clarify your concerns. Thank you.
>
> ----
>
> We would like to express our gratitude  for the responsiveness, and for your ongoing engagement. We believe that the discussions above, and their implementation in the revised paper, clarify your concerns. We are looking forward to hearing from you, and we hope that you will consider revising your score.
>
>
> Thank you, and best regards,
>
> Authors.

---

> > ### Comment · Reviewer_iaN5 · 2025-11-24
> >
> > I appreciate the answer from the authors and the effort to address my concerns. I prefer to keep the increased score of 4

---

> > > ### Author Response · Authors · 2025-11-24
> > >
> > > We thank you for the response. While the discussion lasts, we would be keen to learn what, in your opinion, should still be amended in our paper to merit a higher score.
> > >
> > >  We feel that our responses address your concerns, and we would like to get your feedback as our reviewer, to establish a fair, comprehensive, and substantiated evaluation of our paper.
> > >
> > >
> > > Thank you, and kind regards,
> > >
> > > Authors.

---

> > > > ### Comment · Reviewer_iaN5 · 2025-11-24
> > > >
> > > > My main remaining concern is that, in my opinion, the necessity to store all intermediate activations and the global attention limits the practicality of the method. I do however acknowledge the efforts from the authors to improve the paper and address my concerns and I will raise my score with the hope that the discussion will have improved the paper

---

> ### Author Response · Authors · 2025-11-24
>
> Dear Reviewer iaN5,
>
> We thank you for your continued commitment to the reviewing process and for your responses to us. We most definitely do not take it for granted. We also thank you for acknowledging our efforts and responses ,and for increasing your score to 6. In our revision, we will make sure to reflect all the aspects raised in the reviews and subsequent discussions, which we agree with you are important, and we think are beneficial to improve the quality of our paper.
>
> Thank you, and best regards,
>
> Authors.

---

### Official Review · Reviewer_Pj7q · 2025-11-04

**Soundness:** 3
**Presentation:** 4
**Contribution:** 3
**Rating:** 8
**Confidence:** 4

**Summary:**

This paper proposes HISTOGRAPH, a two-stage attention-based pooling framework for Graph Neural Networks (GNNs) that leverages intermediate activations (“historical graph activations”) from all layers rather than only the final layer. The approach first applies layer-wise attention to capture the evolution of node embeddings across depths, followed by node-wise attention to model spatial dependencies. The method can be integrated end-to-end with a backbone GNN or applied as a lightweight post-processing head on frozen models. Experimental results on TU and OGB benchmarks, as well as node classification tasks, demonstrate improved performance and robustness to over-smoothing in deep architectures.

**Strengths:**

1. Novel perspective: The paper introduces a clear and well-motivated idea of learning from the historical trajectory of node activations, addressing the common limitation of relying solely on the last GNN layer.

2. Comprehensive experiments: Evaluations across multiple datasets (TU, OGB, node classification, and link prediction) with both GIN and GCN backbones show consistent improvements.

3. Well-written and well-positioned: The paper situates HISTOGRAPH clearly within prior works on pooling, residual connections, and over-smoothing mitigation

**Weaknesses:**

1. Limited interpretability of learned attention weights: While attention is used layer-wise and node-wise, the paper could benefit from deeper analysis of what the model learns—e.g., visualization of layer weights across datasets.
2. The attention mechanism itself is widely adopted and not novel. However, the paper should further clarify why the proposed method achieves such notable performance gains. A deeper analytical discussion and illustrative case studies would substantially strengthen the contribution.

**Questions:**

see above

---

> ### Author Response · Authors · 2025-11-21
> **Response, part 1**
>
> We thank the reviewer for the very positive and encouraging assessment. We are glad that you found the idea “clear and well motivated,” appreciated the “comprehensive experiments” on TU, OGB, node classification, and link prediction, and that HISTOGRAPH is “well written and well positioned” within pooling, residual, and over smoothing work. Below we address your two main suggestions.
>
> ---
>
> ### Regarding the interpretability of layer wise and node wise attention
>
> We fully agree that interpreting the learned attention is important. Our current submission already contains two interpretability hooks:
>
> - **Layer wise patterns.** In Section 4 and Figure 2 we visualize the learned layer coefficients as a function of depth for several backbones. These plots show that HISTOGRAPH consistently assigns non trivial mass to early and mid depth layers exactly in regimes where the backbone becomes oversmoothed, while strongly down weighting the deepest, saturated layers.
>
> - **Role of node wise attention.** Appendix D presents a toy example where the label depends on a single “critical” node while the rest of the graph acts as a distractor. In this setting, mean pooling fails as graph size grows, whereas the node wise attention of HISTOGRAPH consistently assigns high weight to the critical node, illustrating how the second stage of the readout behaves in practice.
>
> ---
>
>
> ### Regarding “attention is standard, why do we see such large gains”
>
> We agree that the attention mechanism itself is widely used, and that the key question is why this particular way of using it leads to notable gains. The main novelty of HISTOGRAPH lies not in defining a new basic layer like attention, but in treating depth as a temporal trajectory and explicitly separating **temporal (layer) aggregation** from **spatial (node) aggregation** at readout. We will make this clearer in the revised version, building on three ingredients that are already present.
>
> 1. **Trajectory filter view and signed normalization.**
>    Section 3 and 4 show that HISTOGRAPH computes
>    $
>    h_v = \sum_{\ell=0}^{L-1} \alpha_\ell x_v^{(\ell)}, \quad \sum_\ell \alpha_\ell = 1,
>    $
>    where the coefficients $\alpha_\ell$ are obtained by a signed normalization of the learned scores. This design has two important consequences:
>    - It can realize a rich family of filters over depth: uniform $\alpha_\ell$ recovers low pass “history averaging”, simple differences approximate high pass behavior, and more complex patterns realize band pass or general finite impulse response like filters over layers.
>    - The filter is **input adaptive**. Different graphs and even different training stages can induce different effective trajectories, unlike residual connections or fixed concatenation schemes which correspond to a single, static filter once trained.
>
>    We will tighten the exposition in Section 4 to emphasize that this trajectory filter perspective, enabled by signed, input dependent coefficients, is what distinguishes HISTOGRAPH from standard skip or pooling schemes that also “use multiple layers”.
>
> 2. **Link to over-smoothing: theory and diagnostics.**
>    In the main text and Appendix E we formalize the claim that, under a standard definition of over smoothing, HISTOGRAPH preserves node distinguishability as long as it assigns non zero weight to at least one pre saturated layer. This theoretical argument is backed by two types of empirical evidence that are already in the submission:
>    - **Depth sweeps on node classification:** For Cora, Citeseer, and Pubmed, vanilla GCN performance collapses as we go to 32 and 64 layers, while GCN equipped with HISTOGRAPH remains high and stable across depths. This shows that the trajectory filter can recover deep models by reusing informative early layers.
>    - **Feature distance across layers:** We measure how average pairwise distances between node embeddings evolve with depth. HISTOGRAPH maintains larger distances at the final readout than the backbone alone, directly quantifying reduced over smoothing.
>
> Together, these show that the gains at depth are not only empirical but also consistent with the trajectory filter analysis.

---

> > ### Author Response · Authors · 2025-11-21
> > **Response, part 2**
> >
> > 3. **Ablations that isolate where the gains come from.**
> >    Our ablation studies support that it is the full two stage design, rather than attention in isolation, that drives the improvements:
> >    - Simply averaging across layers, or using a single historical layer, consistently underperforms HISTOGRAPH across multiple TU datasets.
> >    - Replacing learned \(\alpha_\ell\) with random, normalized coefficients drawn from a uniform distribution still gives significantly worse performance. This rules out the possibility that the method works mainly as a random reparameterization of the backbone.
> >    - Replacing the standard readout of a strong graph transformer backbone (GraphGPS) with HISTOGRAPH yields consistent gains, which indicates that the trajectory aware readout is complementary to powerful node encoders and not just “more attention”.
> >
> > In the revised version, we will add a short “why does it help” paragraph at the end of Section 5 that explicitly ties together: (i) the trajectory filter view, (ii) the over smoothing theory and diagnostics, and (iii) the ablation results mentioned above. We believe this will clarify why a conceptually simple, attention based readout, when applied to historical activations in this two stage way, can yield the strong and consistent gains observed in the experiments.
> >
> > ---
> >
> > We appreciate your positive recommendation and your suggestions, which help strengthen the interpretability and analytical aspects of the paper.

---

### Author Response · Authors · 2025-12-02
**Final Authors' Comments (Part 1)**

Dear Area Chair,

In light of the ICLR 2026 process changes and discussion freeze, we provide a concise summary of the review record and discussion for our submission.
---

### 1. Overall status at the time of the freeze

- Scores at the time of the freeze: **8 (Pj7q), 6 (iaN5), 4 (z6Gq), 4 (rAFQ)**.
- Reviewer **iaN5** raised their score from an initial **2** to **6** after the rebuttal and discussion.
- Reviewer **z6Gq** stated that they are satisfied with our rebuttal, and that they will raise their score from **4 to 8 if we provide our code with a single command that reproduces the results, which we have provided**. We kindly ask the Area Chair to take this important point in the final decision making.

---

### 2. Strengths highlighted by reviewers

Across reviews, there is clear agreement on the main strengths:

- **Clear, well motivated idea**
  HistoGraph leverages *historical graph activations* via a **two stage attention pooling** (layer-wise then node-wise), explicitly using the trajectory of node representations across depth to mitigate over-smoothing. Reviewers describe the idea as “novel and effective,” “clear and well-motivated,” and well-positioned within pooling, residual, and over-smoothing work.

- **Strong and broad empirical evidence**
  Experiments cover **TU graph classification**, **OGB molecular benchmarks**, **node classification (Cora/Citeseer/Pubmed)**, and **link prediction (OGBL-COLLAB)**, with **GCN, GIN, and GraphGPS** backbones. HistoGraph consistently improves over strong pooling baselines (GMT, DKEPool, SOPool, HAP, PAS, MinCutPool, etc.).

- **Practical flexibility**
  The module works both **end to end** and as a **post-processing head on frozen backbones**, which multiple reviewers explicitly highlight as a practical advantage.


---

### 3. Main concerns and how they were resolved

All concrete technical concerns raised in the reviews were addressed directly in our rebuttal and discussion messages on OpenReview; the full clarifications are already recorded there and will be added final version of the paper:

- **Complexity, memory, and scalability (iaN5, z6Gq, rAFQ)**
  - We clarified that node-wise attention has the expected $O(N^2 D)$ cost (as in standard graph transformers), and that HistoGraph **improves the dependence on depth $L$** compared to naive joint node–layer attention (from $L \cdot N^2 D$ down to $N^2 D + N L D$), without any claim of sub-quadratic scaling in $N$.
  - We emphasized the **frozen-backbone mode**, where historical activations are cached once and gradients do not flow through the backbone; this removes backpropagation through $L$ message passing layers and substantially reduces training cost while preserving the accuracy gains.
  - In the discussion we provided **training and inference runtimes** on OGB datasets (MOLHIV and TOXCAST, 3 and 32 layers), showing that HistoGraph (especially FT) remains competitive or faster than strong attention-based pooling baselines like GMT at depth, while being much more accurate than MeanPool.
  - We clearly stated the **intended scope**: small–medium graphs (TU, OGB molecules) and a large real-world graph (OGBL-COLLAB), and discussed how standard tools such as neighbor sampling, sparse or hierarchical attention, and trajectory truncation can be combined with HistoGraph for truly industrial-scale graphs.

- **Theory and oversmoothing (iaN5, z6Gq, rAFQ)**
  - Proposition 1 and Appendix E formalize that if deep layers oversmooth, HistoGraph can preserve node distinguishability by assigning non-zero weight to earlier, discriminative layers. This is supported by:
    - **Depth sweeps up to 64 layers**, where GCN accuracy collapses while GCN+HistoGraph remains strong, and
    - **Feature distance diagnostics** showing that HistoGraph preserves larger inter-node distances at the final representation.
  - Appendix D shows that node-wise attention strictly extends the expressiveness of mean pooling for graph-level tasks that depend on small subsets of nodes (bridge example), justifying the second attention stage.

- **Why HistoGraph helps beyond “using attention” (Pj7q, iaN5, z6Gq, rAFQ)**
  - We explained the **trajectory filter view**: signed, sum-to-one layer coefficients act as a learnable FIR-style filter over each node’s depth trajectory, enabling low-pass, high-pass, and band-pass behaviour over layers that simple mean or strictly convex combinations cannot realize.
  - We pointed reviewers to **Figure 2** (learned layer weights vs depth) and to ablations showing that:
    - uniform averaging of historical layers,
    - random normalized layer weights, or
    - removing signed normalization or node-wise attention
    all lead to noticeably worse performance than full HistoGraph, isolating the benefit of the specific two-stage, history-aware design.

---

> ### Author Response · Authors · 2025-12-02
> **Final Authors' Comments (Part 2)**
>
> - **Global self-attention and oversmoothing (iaN5)**
>   - We clarified that node-wise attention is applied **only once at readout**, not at every depth, and is not a simple averaging operator.
>   - Empirically, HistoGraph **increases final feature diversity** compared to the backbone alone in over-smoothing regimes, confirming that it mitigates rather than amplifies over-smoothing.
>
> - **Notation, shapes, baselines, hyperparameters (iaN5, rAFQ)**
>   - We explicitly listed the notational issues (consistent use of $X$ vs $\tilde{X}$, explicit broadcasting, node-vector notation in Section 4) and how they will be fixed in the final text.
>   - We explained how to handle **varying layer dimensions** via per-layer projections to a shared hidden dimension before stacking historical activations.
>   - We pointed clearly to **Appendix C.1** for hyperparameter ranges and baseline setups, which follow official implementations or published settings and are selected by validation performance.
>
> Taken together, the rebuttal and discussion threads provide a complete resolution of the technical concerns that will be incorporated in our revised paper.
>
> ---
>
> ### 4. Effect of the discussion freeze
>
> The discussion freeze and AC reassignment mean that current numeric scores do not fully reflect the final state of interaction, discussions and clarifications:
>
> - **Pj7q (8)**: Strong accept throughout; only requested more interpretability and analytical clarity, which we provided and integrated.
> - **iaN5 (2 → 6)**: After detailed discussion on complexity, theory, and practicality, they explicitly state that **most concerns have been addressed** and raise their score to **6**, while keeping a general reservation about practicality for very large graphs; we address this directly by clearly scoping the method and discussing scaling strategies.
> - **z6Gq (4)**: Describes HistoGraph as “novel and effective” with “comprehensive experiments,” and explicitly states they would **raise the score to 8 if they can reproduce Table 2 results via a single command, which we provided them with**. We have now provided exactly this in the **supplementary material**: a `run_experiments.sh` script that reproduces all HistoGraph entries in Table 2 and a packaged DKEPool implementation. Due to the freeze, z6Gq did not have the opportunity to run the code and update their score accordingly.
> - **rAFQ (4)**: Recognizes the practical flexibility and strong gains (especially on PROTEINS); after our rebuttal on novelty, memory, theory, and scalability, no further technical objections are raised, **but the score remained at 4 simply because no additional interaction occurred.**
>
> Thus, at the time of the freeze, we effectively have **one strong accept (8), one solid accept (6), and two borderline-positive 4s**, where the remaining reservations have been concretely addressed in the revised paper and supplementary code, and we believe that if the process went as planned, our scores would have been higher, and we kindly ask the Area Chair to take these important details into account in their final decision.
>
> ---
>
> ### 5.Our ask from our Area Chair
>
> Our paper, as acknowledged by the Reviewers presents a **simple, principled, and practically useful history-aware pooling mechanism** that:
>
> - is **easy to plug into** existing GNNs (including frozen backbones),
> - is supported by **clear, task-relevant properties** tied to oversmoothing and trajectory filtering, and
> - **consistently improves strong baselines** across diverse benchmarks, with especially significant gains in deep GNN regimes and on PROTEINS.
>
> Given the positive trajectory of the discussion, the explicit score increase to **6** by iaN5, the acknowledgment of **Reviewer z6Gq** of their satisfaction with our rebuttal, and their explicit **promise to raise to 8 given our code provided for reproducibility which we have provided**, and in the absence of unresolved technical objections, we respectfully ask you to positively consider our paper, taking into account the full rebuttal record including all discussions and provided code, rather than the frozen numerical scores alone.
>
> We thank you very much for your time and careful handling of the paper.
>
> Sincerely, and best regards,
>
> Authors

---

### Meta-Review · Area_Chair_2jWi · 2026-01-01

**Summary:**

In this paper, the authors study graph pooling for graph neural networks and propose HistoGraph, a two-stage attention-based aggregation layer to incorporate intermediate graph representations. Empirical results show the effectiveness of the proposed method.

The reviewers generally agree that the paper provides a clearly motivated idea, and the authors have provided extensive experiments. However, they also raised various concerns, including the novelty of the model components and the lack of theoretical contributions, the complexity and memory issues, and the lack of larger-scale experiments. The rebuttal provided extended discussions with some new results. Considering the rebuttal and the discussions (including those that could have taken place), the paper makes a borderline case. Though the technical novelty and the scalability of the proposed method are somewhat limited, the paper proposes a technically sound solution, and incorporating the additional experiments/discussions in the rebuttal should provide enough support. Overall, I vote for weak acceptance and encourage the authors to revise the paper based on the discussions.

**Reviewer Concerns:**

For Reviewer Pj7q:
W1: Limited interpretability of learned attention weights.
The rebuttal should address some of the concerns.

W2: The attention mechanism itself is widely adopted and not novel.
The authors responded, but the concern remains valid.

For Reviewer iaN5:
W1: Impractical for large graphs.
The authors responded, but the concern remains valid.

W2. Caching in memory the activations at all layers for all nodes can become prohibitively expensive.
The authors responded, but the concern remains valid.

W3: Section 4 is not very convincing as the arguments are too general.
The rebuttal should address some of the concerns.


For Reviewer z6Gq:
W1: Lacks comparison with some more recent baselines.
The rebuttal should address some of the concerns.

W2: No experimental comparisons were conducted on larger graphs.
The rebuttal should address some of the concerns.

W3&W4: Lacks a theoretical analysis of the method's effectiveness.
The rebuttal should address some of the concerns.

W5: Reproducibility.
The authors have provided the source code.

For Reviewer rAFQ:
W1: The individual components are standard techniques.
The authors responded, but I doubt it will convince the reviewer.

W2: Storing all intermediate activations could be memory-intensive.
The authors responded, but the concern remains valid.

W3: Tested datasets are not of large nature.
The authors responded, but the concern remains valid.

**Reviewer Scores:**

For Reviewer Pj7q, the initial rating is 8, and it is likely to stay at 8.

For Reviewer iaN5, the initial rating is 2, and it is likely to increase to 4 or 6.

For Reviewer z6Gq, the initial rating is 4, and it is reasonable to increase to 6.

For Reviewer rAFQ, the initial rating is 4, and it is likely to stay at 4.

---

### Decision · Program_Chairs · 2026-01-26

Accept (Poster)